# Validation of lipid-related therapeutic targets for coronary heart disease prevention using human genetics

María Gordillo-Marañón [1✉], Magdalena Zwierzyna [1,2], Pimphen Charoen[1,3,4], Fotios Drenos[1,5], Sandesh Chopade[1,2], Tina Shah[1,2], Jorgen Engmann [1,2], Nishi Chaturvedi [1,6], Olia Papacosta[7], Goya Wannamethee[7], Andrew Wong [6], Reecha Sofat[8], Mika Kivimaki [9], Jackie F. Price[10], Alun D. Hughes [1,2,6], Tom R. Gaunt [11,12,13], Deborah A. Lawlor [11,12,13], Anna Gaulton [14], Aroon D. Hingorani [1,2,16], Amand F. Schmidt [1,2,15,16] & Chris Finan [1,2,15,16]

Drug target Mendelian randomization (MR) studies use DNA sequence variants in or near a gene encoding a drug target, that alter the target's expression or function, as a tool to anticipate the effect of drug action on the same target. Here we apply MR to prioritize drug targets for their causal relevance for coronary heart disease (CHD). The targets are further prioritized using independent replication, co-localization, protein expression profiles and data from the British National Formulary and clinicaltrials.gov. Out of the 341 drug targets identified through their association with blood lipids (HDL-C, LDL-C and triglycerides), we robustly prioritize 30 targets that might elicit beneficial effects in the prevention or treatment of CHD, including NPC1L1 and PCSK9, the targets of drugs used in CHD prevention. We discuss how this approach can be generalized to other targets, disease biomarkers and endpoints to help prioritize and validate targets during the drug development process.

[1] Institute of Cardiovascular Science, Faculty of Population Health, University College London, London WC1E 6BT, UK. [2] UCL British Heart Foundation Research Accelerator, London, UK. [3] Department of Tropical Hygiene, Faculty of Tropical Medicine, Mahidol University, Bangkok 10400, Thailand. [4] Integrative Computational BioScience (ICBS) Center, Mahidol University, Bangkok 10400, Thailand. [5] Department of Life Sciences, College of Health, Medicine, and Life Sciences, Brunel University London, Uxbridge, UK. [6] MRC Unit for Lifelong Health and Ageing, University College London, London WC1E 7HB, UK. [7] Primary Care and Population Health, University College London, London NW3 2PF, UK. [8] Institute of Health Informatics, University College London, London WC1E 6BT, UK. [9] Department of Epidemiology and Public Health, University College London, London WC1E 6BT, UK. [10] Usher Institute, University of Edinburgh, Edinburgh EH8 9AG, UK. [11] MRC Integrative Epidemiology Unit at the University of Bristol, Bristol BS8 2BN, UK. [12] Population Health, Bristol Medical School, University of Bristol, Bristol BS8 2PS, UK. [13] Bristol NIHR Bristol Biomedical Research Centre, University Hospitals Bristol National Health Service Foundation Trust and University of Bristol, Bristol BS8 2BN, UK. [14] European Molecular Biology Laboratory, European Bioinformatics Institute (EMBL-EBI), Wellcome Genome Campus, Hinxton, Cambridge CB10 1SD, UK. [15] Department of Cardiology, Division Heart and Lungs, University Medical Center Utrecht, Heidelberglaan 100, 3584 CX Utrecht, The Netherlands. [16] These authors contributed equally: Aroon D. Hingorani, Amand F. Schmidt, Chris Finan. ✉email: maria.maranon.16@ucl.ac.uk

Genome-wide association studies (GWAS) in patients and populations test relationships between natural sequence variation (genotype) and disease risk factors, biomarkers, and clinical endpoints using population-based cohort and case–control studies.

A well-established role of Mendelian randomization (MR) analysis is to use genetic variants (mostly identified from GWAS) as instrumental variables to identify which disease biomarkers (e.g., blood lipids such as low- and high-density lipoprotein cholesterol (HDL-C) and triglycerides (TG)) may be causally related to disease endpoints (e.g., coronary heart disease; CHD)[1,2]. We and others have shown that variants in a gene encoding a specific drug target, that alters the target's expression or function, can be used as a tool to anticipate the effect of drug action on the same target. We have referred to this application of MR as "drug target MR"[3]. In contrast to a genome-wide biomarker MR, where the variants comprising the genetic instrument are selected from across the genome, in a drug target MR analysis, variants are selected from the gene of interest or neighboring genomic region because these variants are most likely to associate with the expression or function of the encoded protein (acting in *cis*). Whereas genome-wide biomarker MR helps infer the causal relevance of a biomarker for a disease, a drug target MR helps infer whether and, in certain cases in what direction, a drug that acts on the encoded protein, whether an antagonist, agonist, activator, or inhibitor, will alter disease risk (Supplementary Table 1).

Genome-wide biomarker MR studies have validated the causal role of elevated low-density lipoprotein cholesterol (LDL-C) on CHD risk, supporting the findings from randomized controlled trials of different LDL-C lowering drug classes[4–9]. However, such studies have been equivocal on the role of HDL-C and TG in CHD[4,5]. Clinical trials of these lipid fractions have also been seemingly contradictory. For example, using niacin to raise HDL-C did not reduce CHD risk[10], but inhibiting cholesteryl ester transfer protein (CETP) with anacetrapib, which also raises HDL-C, was effective in preventing CHD events[11]. However, a drug target MR of *CETP* on CHD, using variants in the *CETP* gene weighted by their effect on HDL-C, indicates protection from disease (odds ratio (OR): 0.87; 95% CI: 0.84–0.90)[3]. The finding is consistent with the effect of allocation to the CETP-inhibitor anacetrapib in a placebo-controlled trial (0.93; 95% CI: 0.86–0.99) and is compatible with the view that targeting CETP is an effective therapeutic approach to prevent CHD (Fig. 1)[11]. Importantly, as discussed in detail elsewhere[3], drug target MR analyses which use genetic associations with "biomarkers" downstream to the protein, such as HDL-C, use this effect as a proxy for protein concentration or activity (where this has not

been measured directly) and do not provide evidence on whether the biomarker used for the weighting itself mediates disease. Rather, they inform on the validity of the drug target for a disease, regardless of the mediating pathway.

Taken together, these observations suggest that other similarly effective as yet unexploited drug targets might exist for the prevention or treatment of CHD that could be identified through their association with blood lipids even though such analyses should not presume that the effect on CHD is mediated through these lipids.

Here, we apply drug target MR on a set of druggable proteins identified through genetic associations with circulating blood lipids and assessed their causal relevance for CHD. To place the findings in context, we first re-evaluate causal effect estimates for LDL-C, HDL-C, and TG on CHD using "genome-wide biomarker MR", based on summary statistics from GWAS of blood lipids and CHD. Next, we use these data to select genes associated with blood lipids that encode druggable targets and test the effects of these drug targets on CHD using "drug target MR" in two independent datasets. In parallel, we investigate if the genetic associations with each lipid sub-fraction and CHD are consistent with a shared causal variant using genetic co-localization. For a set of replicated, prioritized drug targets, we perform a phenome-wide scan of genetic associations of variants within the encoding gene region with additional disease biomarkers and endpoints beyond CHD. We source data from clinicaltrials.gov and the British National Formulary (BNF) for drugs in clinical phase development and approved medicines, respectively, to identify agents that might be pursued rapidly in clinical phase testing for treatment or prevention of CHD. Because of interest in this area, though not the focus of the work, we also evaluate potential mediators of these effects using multivariable MR (MVMR). Finally, we discuss how this approach might be generalized to other drug targets and clinical endpoints, providing a route to translating findings from GWAS into new drug development.

## Results

**Genome-wide biomarker MR analysis.** Previous genome-wide biomarker MR studies have shown a causal effect of LDL-C and TG on CHD risk, while the causal role of HDL-C remains uncertain[5]. As an initial step, to confirm the robustness of our analytical pipeline and contextualize further analyses, we first replicated previously reported genome-wide biomarker MR estimates using genetic variants from the Global Lipid Genetic Consortium (GLGC)[12] to instrument causal effects of the three lipid sub-fractions on CHD, using summary statistics from the CardiogramPlusC4D Consortium GWAS[13]. Causal estimates were obtained through univariable MR, with Egger horizontal

| | CHD GWAS | Clinical trial | | OR (95% CI) of CHD for intervention or per 1-SD increase in HDL-C |
|---|---|---|---|---|
| **Study** | Events/Total | **Treatment** Events/Total | **Placebo** Events/Total | |
| HDL-C genetic variants (genome-wide) | 60801/184305 | - | - | 0.93 (0.85, 1.02) |
| *CETP* genetic variants | 60801/184305 | - | - | 0.87 (0.84, 0.90) |
| Anacetrapib (CETP inhibitor) | - | 1640/15225 | 1803/15224 | 0.93 (0.86, 0.99) |

**Fig. 1 HDL-C, CETP inhibitor, and CHD: genome-wide biomarker vs. drug target MR.** Forest plot of the HDL-C biomarker MR estimate (Holmes et al., 2015), drug target MR estimate of CETP level and function using HDL-C as a proxy (Schmidt et al., 2020), and odds ratio of anacetrapib clinical trial (HPS3/TIMI55–REVEAL Collaborative Group, 2017). OR odds ratio, CI confidence interval, SD standard deviation.

pleiotropy correction applied through a model selection framework[14]. The OR for CHD per standard deviation (SD) higher concentration of the corresponding blood lipid fraction was 1.50 (95% confidence interval (CI): 1.39–1.63) for LDL-C, 0.95 (95% CI: 0.90–1.01) for HDL-C and 1.10 (95% CI: 1.01–1.21) for TG. These findings were replicated in an independent analysis using summary statistics from a GWAS meta-analysis of lipids measured using a nuclear magnetic resonance (NMR) spectroscopy platform[15,16], and genetic associations with CHD risk derived from UK Biobank[17]. The OR for CHD per SD increase in LDL-C and TG in the replication dataset were 1.28 (95% CI: 1.25–1.31) and 1.23 (95% CI: 1.14–1.32), respectively, and 0.89 (95% CI: 0.83–0.96) per SD increase in HDL-C. This genome-wide biomarker MR estimates confirmed the previously reported causal effect of LDL-C and TG on CHD risk but illustrate the equivocal role of HDL-C. To account for the correlation between the lipid fractions and evaluate their direct independent effect on CHD, we performed an MVMR analysis in the discovery datasets, which assessed genetic associations with the three lipid subfractions and CHD risk in a single model. The MVMR analysis generated an OR of 1.53 (95% CI: 1.44–1.62) per SD higher LDL-C, 0.91 (95% CI: 0.86–0.95) per SD higher HDL-C, and 1.09 (95% CI: 1.01–1.17) per SD higher TG (Supplementary Table 2).

**Drug target MR analysis**. Drug target MR was used to determine the effect on CHD of perturbing druggable proteins that influence one or more of the three lipid fractions. First, genes previously shown to encode druggable proteins were selected in regions around variants associated with one or more of the major circulating lipid subfractions applying a $P$ value $< 1 \times 10^{-6}$. This identified 341 genes; 149 for an association with LDL-C, 180 for HDL-C, and 154 for TG[18]. One hundred forty genes (41%) were associated with a single lipid subfraction, 101 (30%) were associated with two subfractions and 100 (29%) were associated with all three subfractions (Supplementary Fig. 1, Supplementary Data 1). Subsequently, we performed a drug target MR analysis on CHD accounting for genetic correlation between variants (see "Methods"). In the absence of direct measures of the encoded protein, we proxied the effect of genetic drug target perturbation through the downstream effect on one or more of the three lipid sub-fractions. Here, we used genetic associations with LDL-C, HDL-C, and TG as a proxy for drug target effects on CHD, which does not provide direct evidence on whether the drug target itself affects CHD through the leveraged lipid weight; this mediation question is subsequently explored using MVMR.

Of the 341 drug targets, 165 could be associated with CHD, with 131 of these estimates being consistent with a protective effect when instrumented for a reduction in LDL-C or TG and/or elevation in HDL-C (Fig. 2, Supplementary Data 2). When weighted by LDL-C, eighty-seven targets showed a significant effect on CHD after orientating towards an increasing LDL-C direction, with the first and third quartiles (Q) of the CHD OR of 1.93 and 3.32. Similarly, the Q1 and Q3 after orientating the OR toward an increasing HDL-C direction were 0.22 and 0.53 for the 49 significant HDL-C instrumented targets, and for the 49 significant TG instrumented targets these were 1.95 and 4.35, respectively.

To assess the potential for false-positive results, the distribution of the exposure-specific $P$ values was tested against the uniform distribution expected under the null hypothesis[19]. The Kolmogorov–Smirnov (KS) goodness-of-fit test was not consistent with the hypothesis that the observed findings could be readily explained by multiple testing (Supplementary Fig. 2).

**Rediscoveries of indications and on-target adverse effects**. We investigated if the drug target MR analysis rediscovered the

mechanism of action of drugs with a license for lipid modification or compounds with a different indication but with reported lipid-related effects. To do so, compounds with reported lipid indications or adverse effects were extracted from the BNF website (https://bnf.nice.org.uk/), which comprises prescribing information for all UK licensed drugs. Out of the 341 druggable genes included in the analysis, five encoded the targets of drugs with a lipid-modifying indication (PCSK9, PPARG, PPARA, NPC1L1, and HMGCR) of which NPC1L1, HMGCR, and PCSK9 are targets of drugs used in CHD prevention; and six encoded a protein target of a drug with reported lipid-related adverse effects (ADRB1, TNF, ESR1, FRK, BLK, and DHODH) (Supplementary Data 3). To include outcome and side effect data of candidates in clinical phase development, the 341 drug targets were mapped to compound data available in a clinicaltrials.gov curated database. This database differentiates between endpoints monitored throughout the trial ("outcomes") and unanticipated harmful episodes during the study that may be on-target or off-target effects of the trial agent ("adverse events"). Of the 341 drug targets, 23 had reported lipid-related outcomes and 40 had reported lipid-related adverse events (Supplementary Data 3).

The pool of druggable targets that were modeled using higher LDL-C as a proxy for the pharmacological action on a drug target included 14 targets of clinically used drugs, three of which were licensed for CHD treatment by lowering LDL-C (HMGCR, PCSK9, and NPC1L1). The non-CHD indications of clinically used drugs included dyslipidemias (PPARA), type 2 diabetes (PPARG and NDUFA13), autoimmune diseases (TNF), neoplasms (RAF1 and PSMA5), circulatory disorders (ABCA1, PLG, ITGB3, and F2), and alcohol-dependency (ALDH2) (Table 1). With the exception of F2, instrumenting the target action through a higher LDL-C effect was associated with a higher CHD risk. Two drug targets were for compounds already in phase 3 trials for CHD prevention (ANGPTL3 and CETP). Lastly, three targets were in phase 2 trials of compounds developed for other indications (CYP26A1, LTA, and LTB). The remaining 82 of the 101 targets had not yet been drugged by compounds in clinical phase development.

When using higher HDL-C as a proxy for pharmacological action, MR of four drug targets with compounds approved for non-CHD indications showed a directionally beneficial effect on CHD (VEGFA, PSMA5, CACNB1, and NISCH), suggesting potential for indication expansion (Table 1). Three were targets for drugs approved for non-CHD indications but which showed a potentially detrimental effect direction on CHD when instrumented through increasing HDL-C concentration (ESR1, ALOX5, and TUBB). Both CYP26A1 and CETP were associated with lower CHD risk when the effect on CHD was instrumented through an elevation of HDL-C. The remaining 65 of the 74 targets have not yet been drugged by compounds in clinical phase development.

Lastly, the set of druggable targets with compounds developed for non-CHD indications that were modeled using higher TG as a proxy for the pharmacological action on the target included PPARG, DHODH, VEGFA, TOP1, TUBB, NDUFA13, ABCA1, BLK, and F2 (Table 1). Of these, instrumenting the CHD effect through higher TG via drug action on BLK or F2 increased CHD risk. For the remaining targets, which included CETP, ANGPTL3, and CYP26A1, instrumenting the target effect through lowering TG levels decreased the risk of CHD, while the remaining 52 of the 64 targets have not been drugged by licensed compounds or clinical candidates yet.

**Independent validation of the drug target MR estimates**. To help verify the MR findings and reduce the multiple testing burden, an independent two-sample drug target MR analysis was

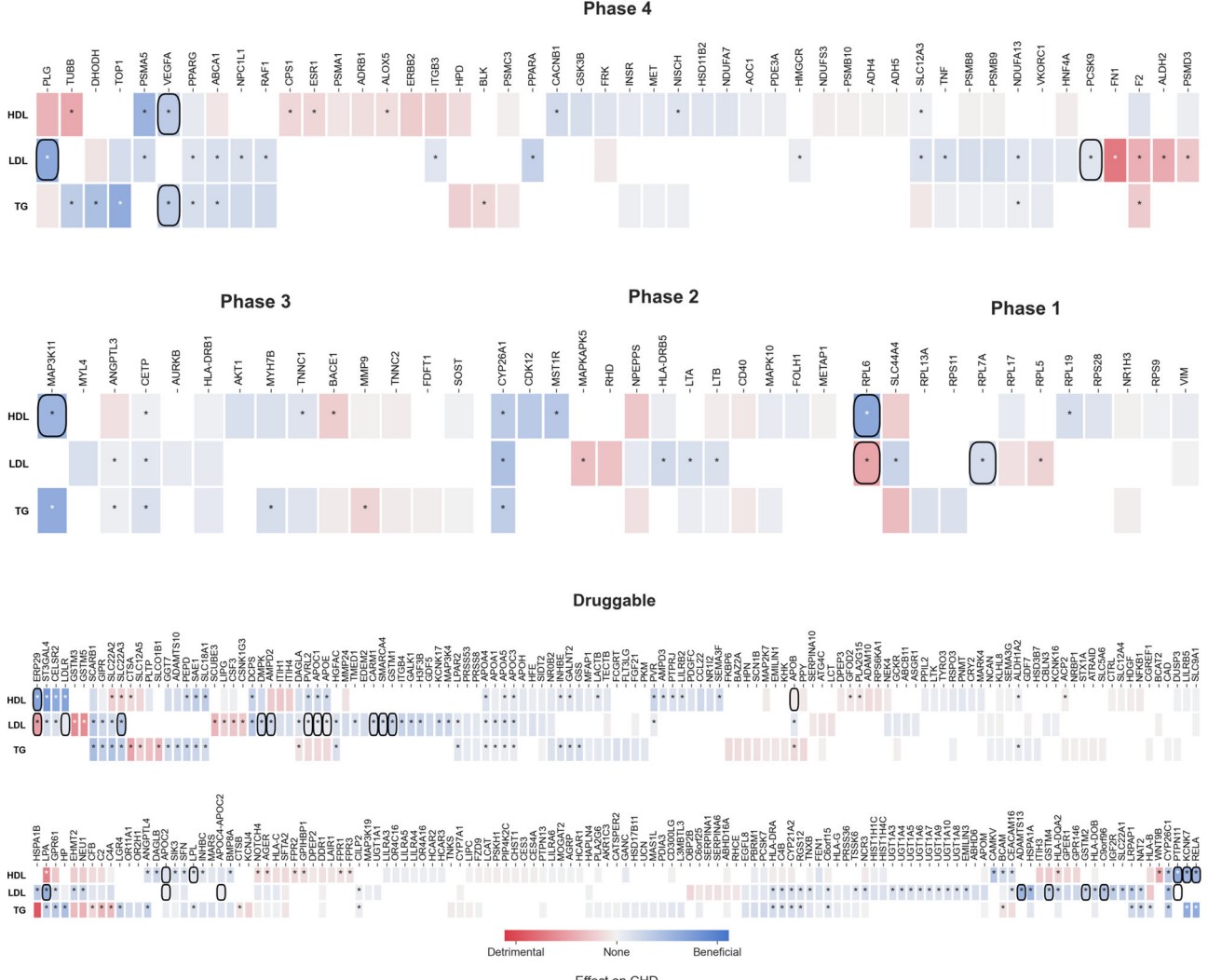

**Fig. 2 Discovery drug target MR estimates on CHD.** Analyses were performed using genetic associations with LDL-C, HDL-C, and TG from the Global Lipid Genetic Consortium (GLGC) with CHD events from the CardiogramPlusC4D Consortium. Drug targets are grouped by clinical phase according to the ChEMBL database. Blue indicates a beneficial effect on CHD risk and red a detrimental effect per SD difference with respect to the indicated lipid sub-fraction. Significant estimates are indicated with an asterisk (*). Co-localization of genetic effects on the relevant lipid sub-fraction and CHD at the same locus is indicated by a square around the cell.

conducted using summary statistics from a GWAS of blood lipids measured using an NMR spectroscopy platform[15,16], and genetic associations with CHD risk derived from UK Biobank[17]. The validation analysis identified 47 significant MR estimates (P value < 0.05), of which 39/47 (83%) showed a concordant direction of effect with the initial analysis (Fig. 3) corresponding to 30 drug targets. Replicated targets included the licensed LDL-lowering drug targets PCSK9 and NPC1L1 (Table 2). While the majority of the replicated drug targets were anticipated to decrease CHD risk when instrumenting their effect through LDL-C concentration based on the univariable results, nine of the drug targets analyzed were significantly associated with lower CHD when the drug target effects were modeled through HDL-C and/or TG (Supplementary Fig. 3).

**Discriminating independent lipid effects using MVMR.** After considering each lipid sub-fraction as a single measure on disease risk in the univariable drug target MR analyses, we performed a multivariable drug target MR (MVMR) analysis including LDL-C, HDL-C, and TG in a single model to account for potential

pleiotropic effects of target perturbation via the other lipid sub-fractions and, in contrast to the previous univariable drug target MR, attempt to directly identify any potential lipid mediating pathway. Twenty-six of the replicated targets had sufficient data (3 or more variants) for the multivariable analysis. This analysis identified a single likely lipid fraction for 12 targets (SLC12A3, APOB, APOA1, PVRL2, APOE, APOC1, CELSR2, GPR61, PCSK9, and CEACAM16 through LDL-C; LPL through HDL-C; and ALDH1A2 through TG) (Supplementary Data 4). We found that SMARCA4 and APOA5 likely affected CHD through LDL-C and TG and that RPL7A likely affected CHD through LDL-C and HDL-C pathways. Due to the limited number of variants in VEGFA, CILP2, NDUFA13, and ANGPTL4, multivariable MR analysis could not distinguish the lipid fraction through which CHD was likely affected. Additionally, the presence of horizontal pleiotropy in the MVMR analysis based on heterogeneity tests suggested that PCSK9, LPL, APOC1, APOE, PVRL2, APOB, APOC3, CETP, APOA1, and CELSR2 may affect CHD through additional pathways beyond the lipid sub-fractions LDL-C, HDL-C, and TG included in the current model.

**Table 1 Univariable drug target MR estimates for drug targets approved for indications other than lipid-lowering.**

| Drug target gene | LDL-C (OR, 95% CI) | HDL-C (OR, 95% CI) | Triglycerides (OR, 95% CI) | Mechanism of action and indication |
|---|---|---|---|---|
| ESR1 | – | 2.11 (1.13–3.93)* | – | AGONIST: Neoplasms, Hypogonadism, Menorrhagia, Primary Ovarian Insufficiency, Acne Vulgaris, Postmenopausal Osteoporosis<br>ANTAGONIST: Breast Neoplasms, Neoplasms<br>MODULATOR: Infertility, Dyspareunia, Breast Neoplasms, Postmenopausal Osteoporosis |
| TNF | 2.03 (1.05–3.93)* | – | 1.21 (0.78–1.9) | INHIBITOR: Ankylosing Spondylitis, Crohn Disease, Psoriasis, Rheumatoid Arthritis, Colitis, Ulcerative, Psoriatic Arthritis, Immune System Diseases, Juvenile Arthritis |
| BLK | – | – | 0.46 (0.31–0.7)* | INHIBITOR: Precursor Cell Lymphoblastic Leukemia–Lymphoma, Neoplasms |
| DHODH | 0.66 (0.44–1.0) | – | 7.42 (2.32–23.71)* | INHIBITOR: Rheumatoid Arthritis, Immune System Diseases, Multiple Sclerosis |
| PPARG | 1.67 (1.04–2.68)* | 0.71 (0.35–1.48) | 2.18 (1.14–4.15)* | AGONIST: Type 2 Diabetes Mellitus, Diabetes Mellitus, Ulcerative Colitis, Cardiovascular Diseases |
| PPARA | 3.77 (1.44–9.85)* | – | – | AGONIST: Cardiovascular Diseases, Hypercholesterolemia, Dyslipidemias |
| NDUFA13 | 1.63 (1.13–2.35)* | – | 1.18 (1.0–1.39)* † | INHIBITOR: Diabetes Mellitus, Type 2 Diabetes Mellitus |
| ALDH2 | 0.14 (0.07–0.29)* | – | – | INHIBITOR: Ectoparasitic Infestations, Alcoholism |
| NISCH | – | 0.57 (0.35–0.93)* | 1.16 (0.31–4.34) | AGONIST: Hypertension |
| ABCA1 | 2.05 (1.34–3.15)* | 1.41 (0.66–3.0) | 2.4 (1.29–4.49)* | INHIBITOR: Cardiovascular Diseases |
| F2 | 0.17 (0.05–0.59)* | 0.57 (0.13–2.43) | 0.35 (0.13–0.94)* | INHIBITOR: Venous Thrombosis, Thrombosis, Unstable Angina, Thrombocytopenia, Atrial Fibrillation, Embolism, Stroke |
| TUBB | – | 7.56 (1.18–48.38)* | 4.46 (2.13–9.36)* | INHIBITOR: Breast Neoplasms, Neoplasms, Hodgkin Disease, Large-Cell Anaplastic Lymphoma, Non-Small-Cell Lung Carcinoma, Gout, Familial Mediterranean Fever |
| VEGFA | – | 0.22 (0.15–0.3)* | 4.16 (2.45–7.08)* † | ANTAGONIST: Retinal Neovascularization<br>INHIBITOR: Diabetic Retinopathy, Retinal Neovascularization, Wet Macular Degeneration, Macular Edema, Colorectal Neoplasms, Neoplasms, Glioblastoma, Renal Cell Carcinoma, Non-Small-Cell Lung Carcinoma, Uterine Cervical Neoplasms |
| RAF1 | 2.06 (1.48–2.86)* | – | 2.63 (0.79–8.83) | INHIBITOR: Neoplasms |
| PSMA5 | 2.47 (1.8–3.39)* † | 0.08 (0.02–0.29)* | – | INHIBITOR: Multiple Myeloma, Neoplasms, Mantle-Cell Lymphoma |
| ALOX5 | – | 1.74 (1.18–2.58)* | – | INHIBITOR: Asthma, Ulcerative Colitis, Rheumatoid Arthritis, Juvenile Arthritis |
| CACNB1 | – | 0.38 (0.2–0.72)* | – | BLOCKER: Cardiovascular Diseases<br>MODULATOR: Fibromyalgia, Seizures, Epilepsy, Neuralgia, Restless Legs Syndrome, Postherpetic Neuralgia |
| PLG | 18.35 (5.47–61.6)* | 5.48 (0.07–456.86) | 0.75 (0.18–3.14) | ACTIVATOR: Thrombosis, Pulmonary Embolism, Stroke, Myocardial Infarction, Heart Failure, Hepatic Veno-Occlusive Disease<br>INHIBITOR: Hemorrhage, Menorrhagia |
| ITGB3 | 1.64 (1.06–2.52)* | 2.79 (0.81–9.62) | – | INHIBITOR: Thrombosis, Unstable Angina |
| TOP1 | 2.3 (0.15–35.62) | – | 16.72 (4.19–66.8)* | INHIBITOR: Neoplasms |

These drug targets showed lipid records in clinicaltrials.gov and/or the British National Formulary (BNF). *OR* odds ratio of CHD per 1-standard deviation increase in LDL-C, HDL-C, or triglycerides; *CI* confidence interval.
*Indicates significance in the discovery analysis.
†Indicates significance in both original and validation study and concordant direction of effect.

**Co-localization between loci for lipids and CHD**. Co-localization analyses are often performed to facilitate the mapping of genetic variants to causal genes in a disease GWAS by assessing whether associations with gene expression and disease outcome share a causal variant. Here, we applied co-localization analysis using blood lipids as an intermediate trait instead of gene expression data as a parallel validation step to assess if the genetic associations with each lipid sub-fraction and CHD were consistent with a shared causal variant[20]. Twenty-eight out of a total of 33 co-localization signals overlapped a significant finding in the discovery MR, which corresponded to 25 genes encoding a drugged or druggable target (Fig. 2). Moreover, 11 of the replicated drug targets showed evidence of co-localization between the lipid sub-fraction and CHD. These included 9 targets replicated for lowering LDL-C levels (SMARCA4, PVLR2, APOE, APOC1, CARM1, RPL7A, ADAMTS13, PCSK9, and C9orf96), one target replicated for raising HDL-C levels (LPL), and one target replicated for lowering TG levels (VEGFA).

**Tissue expression to aid drug target prioritization**. While many tissues are involved in lipid homeostasis, the liver is considered the mechanistic effector organ for many therapeutics targeting lipid metabolism[21]. To investigate if the replicated drug target genes were specifically expressed in the liver or any other particular tissue, we extracted their RNAseq expression profiles from the Human Protein Atlas[22] and calculated two commonly used tissue specificity metrics: the tau and z-scores[23]. Whilst tau summarizes the overall tissue distribution of a given gene and helps to distinguish between broadly expressed housekeeping genes (tau = 0) and tissue-specific genes (tau = 1), z-scores quantify how elevated the expression of a particular gene is in a particular tissue compared to other tissues. Among the 30 replicated genes, 28 had available RNAseq data, of which 15 (54%) showed elevated expression in the liver compared to other tissues (z-score > 1) (Table 2, Supplementary Fig. 4). These genes included the known lipid-lowering drug target genes, PCSK9, and NPC1L1. Furthermore, eight genes were highly specific to the liver as indicated by high tau values (tau > 0.8). Other tissues showing elevated expression of the replicated drug target genes were gastrointestinal tissues such as the small intestine and colon (e.g., APOA4 and APOB) and kidney (SLC12A3). Regarding the expression distribution of the targets, 9 showed tau values below

| Source of data | | | | Direction of effect | | | | |
|---|---|---|---|---|---|---|---|---|
| | **Lipids measures** | **Disease endpoints** | | | **LDL-C** | **HDL-C** | **Triglycerides** | **Overall** |
| **Discovery** | Clinical chemistry (GLGC, N= 188,578) | Research-based case ascertainment (CardiogramPlusC4D, N= 184,305 cases) | | **Concordant** | 21 | 6 | 12 | 39 |
| **Replication** | Nuclear magnetic resonance (NMR) spectroscopy (Kettunen et al, 2016, UCLEB Meta-analysis, N=33,029) | Routine Electronic Health Records (UK Biobank, N=34,541 cases) | | **Discordant** | 4 | 0 | 4 | 8 |

**Fig. 3 Replication of drug target MR findings.** The discovery and replication analyses used different data sources for both exposure and outcome. Totally, 145 replication MR analyses were performed in which the gene boundaries included genetic associations exceeding the pre-specified significance threshold ($P$ value $\leq 1 \times 10^{-4}$).

---

## Table 2 Tissue specificity for replicated genes encoding drug targets.

| Drug target gene | LDL-C (OR, 95% CI) | HDL-C (OR, 95% CI) | Triglycerides (OR, 95% CI) | Tissue specificity index (tau) | Top tissues (Z-score > 1) |
|---|---|---|---|---|---|
| APOA5 | 2.05 (1.4–3.02)* † | 0.72 (0.6–0.87)* † | 1.21 (1.12–1.31)* † | 1.00 | Liver |
| SLC12A3 | 1.94 (1.43–2.63)* | 0.89 (0.86–0.93)* † | 0.75 (0.24–2.33) | 0.98 | Kidney |
| CEACAM16 | 1.66 (1.31–2.11)* † | 0.46 (0.27–0.79)* | 0.56 (0.25–1.27) | 0.98 | Pancreas, tonsil |
| APOC3 | 2.04 (1.72–2.42)* † | 0.67 (0.58–0.78)* | 1.26 (1.12–1.41)* † | 0.95 | Liver |
| APOA4 | 1.51 (1.23–1.86)* | 0.53 (0.38–0.74)* | 1.27 (1.14–1.43)* † | 0.94 | Small intestine, colon, duodenum |
| APOB | 1.5 (1.18–1.9)* † | 1.23 (0.72–2.12) | 0.53 (0.29–0.98)* † | 0.94 | Liver, small intestine |
| APOA1 | 1.88 (1.49–2.36)* † | 0.84 (0.63–1.11) | 1.25 (1.12–1.4)* † | 0.93 | Liver, small intestine |
| NPC1L1 | 2.01 (1.48–2.73)* † | – | 2.56 (0.75–8.68) | 0.92 | Small intestine, colon, duodenum, liver |
| GPR61 | 1.97 (1.56–2.5)* † | 3.02 (0.77–11.91) | 5.14 (1.43–18.48)* | 0.91 | Cerebral cortex, adrenal gland, eye, thyroid gland |
| PCSK9 | 1.6 (1.45–1.77)* † | – | – | 0.87 | Liver, lung, pancreas |
| APOC1 | 1.31 (1.22–1.41)* † | 0.39 (0.25–0.59)* | 0.51 (0.17–1.47) | 0.85 | Liver |
| CETP | 1.49 (1.29–1.72)* | 0.91 (0.87–0.95)* † | 1.98 (1.63–2.4)* † | 0.76 | Lymph node, liver, placenta, spleen |
| ADAMTS13 | 11.18 (4.37–28.59)* † | – | – | 0.72 | Liver |
| CILP2 | 1.19 (1.01–1.39)* | – | 1.18 (1.0–1.39)* † | 0.71 | Testis, gallbladder, ovary, thyroid gland |
| LPL | – | 0.63 (0.49–0.82)* † | 1.68 (1.46–1.92)* † | 0.68 | Adipose tissue, breast, heart muscle, seminal vesicle |
| APOE | 1.3 (1.2–1.41)* † | 0.39 (0.26–0.59)* | 0.5 (0.17–1.45) | 0.58 | Liver, adrenal gland |
| CELSR2 | 1.97 (1.78–2.18)* † | 0.06 (0.04–0.09)* | – | 0.58 | Cerebral cortex, fallopian tube, skin |
| ALDH1A2 | – | 0.89 (0.81–0.99)* | 1.28 (1.07–1.54)* † | 0.55 | Endometrium, blood, cervix, uterine, fallopian tube, eye, seminal vesicle, testis |
| ANGPTL4 | – | 0.48 (0.28–0.83)* † | 3.38 (1.02–11.22)* † | 0.50 | Liver, adipose tissue, breast, cerebral cortex, pancreas |
| PVR | 1.31 (1.12–1.54)* † | 0.32 (0.11–0.91)* | – | 0.45 | Liver, heart muscle |
| NDUFA13 | 1.63 (1.13–2.35)* | – | 1.18 (1.0–1.39)* † | 0.43 | Testis, blood, heart muscle, skeletal muscle |
| CARM1 | 2.27 (1.68–3.05)* † | – | – | 0.38 | Skeletal muscle |
| VEGFA | – | 0.22 (0.15–0.3)* | 4.16 (2.45–7.08)* † | 0.33 | Thyroid gland, endometrium, heart muscle, liver, skeletal muscle, urinary bladder |
| SIK3 | 1.15 (0.57–2.31) | 0.46 (0.29–0.73)* † | 1.08 (0.98–1.18) | 0.27 | Cerebral cortex, ovary, parathyroid gland, testis, thyroid gland |
| TMED1 | 2.06 (1.5–2.83)* † | – | – | 0.26 | Blood, heart muscle, liver, placenta, skeletal muscle |
| PSMA5 | 2.47 (1.8–3.39)* † | 0.08 (0.02–0.29)* | – | 0.23 | Liver, cerebral cortex, kidney, skeletal muscle, thyroid gland |
| SMARCA4 | 2.22 (1.98–2.49)* † | 0.01 (0.0–0.02)* | – | 0.19 | Cerebral cortex, bone marrow, esophagus, skeletal muscle, skin, testis, tonsil |
| RPL7A | 2.29 (1.57–3.36)* † | – | – | 0.19 | Salivary gland, endometrium, lymph node, ovary, pancreas |

The tau value is a measure of tissue specificity with values between 0 and 1, where 1 indicates high specificity for a single tissue. The tissue(s) with the highest expression of the gene is indicated in the top tissue column. *OR* odds ratio of CHD per 1-standard deviation increase in LDL-C, HDL-C, or triglycerides; *CI* confidence interval.
*Indicates significance in the discovery analysis.
†Indicates significance in both original and validation study and concordant direction of effect.

---

0.5, indicating that they are broadly expressed and suggesting that, when developing a drug, the possibility of observing adverse effects increases[24].

**Phenome-wide scan of replicated drug target candidates.** The identification of potential mechanism-based adverse effects of a target represents an important aspect when prioritizing clinical candidates in the drug development pipeline. To explore potential effects of target perturbation on clinical endpoints other than CHD (whether beneficial or adverse), we performed a phenome-wide scan in 102 disease traits of the 30 drug targets replicated via drug target MR (Methods, Fig. 4, Supplementary Figs. 5–33). The 102 traits were agnostically selected and represent the entire spectrum of clinical disease available from the Neale Lab UK Biobank release, with the exclusion of Coronary Artery Disease (ICD10 code: I25), and data from 23 publicly available GWAS with the largest sample sizes for such phenotypes. Besides

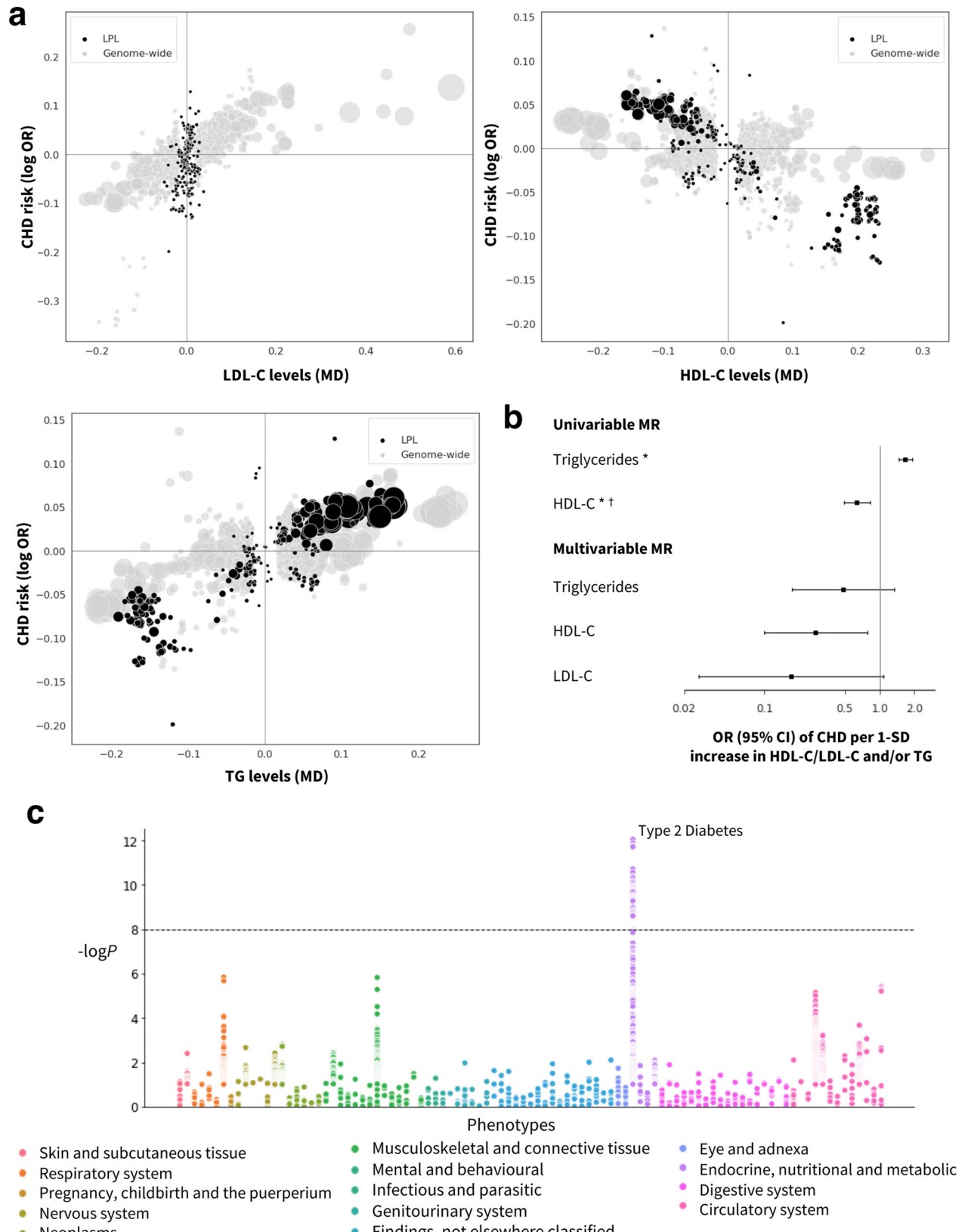

**Fig. 4 Prioritized target: lipoprotein lipase (LPL). a** Genetic associations at the locus (±50 kbp) in black vs. genome-wide associations (gray, P value < 1 × 10$^{-6}$ based on two-sided z-tests). The x-axis shows the per allele effect on the corresponding lipid expressed as mean difference (MD) from GLGC and the y-axis indicates the per allele effect on CHD expressed as log odds ratios (OR) from CardiogramPlusC4D. The marker size indicates the significance of the association with the lipid sub-fraction (P value). **b** Univariable and multivariable (drug target) cis-MR results presented as OR and 95% confidence intervals with lipid exposure (n = 188,577 individuals) and CHD outcome (n = 60,801 cases and 123,504 controls). An asterisk (*) indicates the MR estimates as being replicated, and a dagger (†) that the lipid effect and CHD signals are co-localized. **c.** Disease associations at the locus with 103 clinical endpoints from UK Biobank and GWAS Consortia.

genome-wide significant associations with diseases of the circulatory system, variants in six drug target genes showed genome-wide significant associations with type 2 diabetes (NDUFA13, CILP2, PVRL2, VEGFA, APOC1, and LPL), five with Alzheimer's disease (APOC1, PVR, PVRL2, APOE, and CEACAM16), four with asthma (SMARCA4, CETP, VEGFA, and ALDH1A2) and four with gout (APOA1, APOC3, APOA4, and APOA5). Notably, the PheWAS rediscovered the mechanism of action of metformin, a drug targeting NDUFA13 and licensed for type 2 diabetes[25].

## Discussion

By combining publicly available GWAS datasets on blood lipids and CHD and applying MR approaches with drug information and clinical data, we have genetically validated and prioritized drug targets for CHD prevention.

We identified 131 drug target genes associated with CHD risk from a set of 341 druggable genes overlapping associations with one or more of the major blood lipid fractions. The set of targets included NPC1L1, HMGCR, and PCSK9, which are known targets of LDL-lowering drugs whose efficacy in CHD prevention has been proven in clinical trials. We performed an independent replication study both to corroborate the targets and the direction of the effects. We replicated the findings in independent datasets (UCLEB Consortium and UK Biobank) in which lipids were measured using a different platform (NMR spectroscopy in UCLEB) and the disease endpoints ascertained by linkage to routinely recorded health data (UK Biobank). The validation study replicated 83% (39/47) of the initial estimates, including the mechanism of action of current lipid-modifying drug targets PCSK9 and NPC1L1 and the suggested mechanism of action of compounds under investigation for lipid modification through TG or HDL-C, such as CETP inhibitors[26,27].

As a positive control step, our (genome-wide) biomarker MR analysis replicated previous findings on the potential causal relevance of LDL-C, TG, and HDL-C[5,11,28]. Importantly, contrary to previous studies, here we replicated findings using a completely independent set of NMR-spectroscopy measured lipids data and CHD cases sourced from UK Biobank. While the causal relevance of LDL-C for CHD has been robustly proven through successful drug development of for example statins, there are as yet no compounds licensed for CHD prevention through effects on HDL-C and TG. Hence, the causal relevance of the lipid subfraction, while supported by the current genome-wide biomarker MR analyses, cannot be concluded definitively. It is therefore essential to highlight that, while our drug target analysis uses genetic associations with these lipid sub-fractions as weights, our inference throughout has been on the therapeutic relevance of perturbing the proteins encoded by the corresponding genes which are the main category of a molecular target for drug action. The genetic associations with the corresponding lipids are merely used as a proxy for protein activity and/or concentration, serving to orientate the MR effects in the direction of a therapeutic effect. They do not provide comprehensive evidence on the pathway through which perturbation of such targets causally affects CHD. Nevertheless, MVMR does provide insight on the potential relevance of lipid pathways in mediating the effects of drug target perturbation. In general, results that do not meet the significance threshold should not be over-interpreted as proof of the absence of effect[29]. This may be exacerbated here by potential weak instrument bias, which will be expected to attenuate results towards the no-effect direction.

The set of 30 replicated drug targets also included lipoprotein lipase (LPL), a target that could potentially decrease CHD risk based on the univariable MR findings, with an effect through HDL-C further endorsed by the co-localization and MVMR

analyses (Fig. 4). In contrast to current lipid-lowering drug targets which are specifically expressed in the liver, LPL shows the highest specific expression in adipose tissue which suggests tissues beyond the liver may be relevant to target lipid metabolism. Several pharmacological attempts have been pursued to target LPL[30,31], and gene therapy has also been applied to treat LPL deficiency by introducing extra copies of the functional enzyme in patients with hypertriglyceridemia[32]. The approval of gene therapy interventions and the known indirect activation of LPL by drugs targeting other proteins, such as fibrates[33] and metformin[34], suggest that the previous failure of compounds targeting LPL in initial trials may have been idiosyncratic. LPL activity is also modulated by another protein in the replicated dataset, apolipoprotein A5 (ApoA5), which is exclusively expressed in liver tissue. The MVMR suggests that ApoA5 (partially) affects CHD through LDL-C and TG-mediated pathways. Regardless of the mediating lipid or lipids, the genetic findings in relation to both LPL and ApoA5 are consistent and point to this as an important potentially targetable pathway in atherosclerosis, supporting prior work[35].

To provide an indicative genetic profile of a drug target and hypothesize about potential mechanism-based adverse effects, repurposing opportunities or expansion of the indication portfolio of a drug target, we performed a PheWAS of variants in and around the replicated set of targets on 102 traits. While in some cases PheWAS highlighted associations with particular clinical endpoints, for example, the rediscovery of already known indications or biological pathways, such as the associations of type 2 diabetes with variants in NDUFA13 or the association of Alzheimer's Disease with APOE, further research is needed to evaluate the causal role of the target in the corresponding disease and the beneficial or detrimental effects of modulating those targets pharmacologically.

Some limitations of this study are noteworthy. First, we only included genes regarded as encoding druggable proteins, which currently comprise approximately 25% of all protein-coding genes[18]. As knowledge advances, additional proteins will become druggable, and alternative therapeutic strategies such as antisense oligonucleotides and gene therapy may extend the range of mechanisms that can be targeted. The approach we describe is in fact agnostic to therapeutic modality and could be adapted accordingly. Notably, antisense oligonucleotides were efficiently delivered to the liver[36], where 54% of the prioritized targets in our analysis showed elevated expression compared to other tissues. Second, we assigned variants to druggable genes based on genomic proximity, which may be as reliable as other approaches in mapping causal genes[37–39]. However, simple genomic proximity might result in the misleading assignment of the causal gene in a region containing multiple genes in high LD (e.g., PVRL2, APOC1, and APOE are all located in a region of LD in Chr19:45349432-45422606, GRCh37). In an effort to account for this, all the druggable genes (±50 kbp) that overlap one of the genetic variants associated with LDL-C, HDL-C, or TG were included in the analysis, and we provided information on the proximity of the variant to the gene, a gene distance rank value (in base pairs), and previous gene prioritization data by the Global Lipids Genetics Consortium (GLGC)[12] to inform scenarios in which the causal gene may be a non-druggable gene but reside in the same region (Supplementary Data 1). Lastly, because some but not all of the studies contributing to these consortia measured blood lipids on a fasting sample, we are unable to conduct separate analyses based on genetic effects in the fasting and non-fasting state.

We used cis-MR to evaluate the relevance of each drug target to CHD, which is less prone to violation of the horizontal pleiotropy assumption than MR analyses with trans instruments[3], which

also require direct measurement of the protein of interest. However, cis-MR also requires some decisions to be made regarding instrument selection: defining the locus of interest, the significance threshold for the association with the exposure, and the LD threshold to prune correlated instruments. Since an agreement on the choice of a general LD threshold and flanking region has yet to be reached, we used a window of 50kbp and LD threshold of 0.4, which showed the most consistent estimates in a grid-search in the discovery data using the four positive control examples: PCSK9, NPC1L1, HMGCR, and CETP. Based on previous studies showing that using less stringent $P$ value thresholds often results in improved performance in cis-MR settings, we relaxed the threshold below genome-wide significance to select the genetic associations to instrument the exposure; and accounted for LD correlation by pruning and LD modeling during the MR analysis[3,40].

We addressed multiple testing in the MR analyses in a number of complementary ways. To assess the potential for false-positive results, we tested the distribution of the exposure-specific $P$ values against the uniform distribution expected under the null hypothesis[19]. The KS goodness-of-fit test indicated that the number of extreme $P$ values obtained would be highly unlikely under the null hypothesis, suggesting that they are unlikely to represent false positives. Next, we validated our findings with independent data sources and conducted a second drug target MR, although several drug target genes could not be evaluated in the validation analysis because the gene boundaries did not include genetic associations exceeding the pre-specified significance threshold ($P$ value $\leq 1 \times 10^{-4}$), likely related to the "modest" sample size of the NMR replication data ($N = 33,029$). By drawing inference on replicated data, the multiple testing burden was considerably reduced ($0.05^2 = 0.0025$), which when applied to 98 drug targets retained after replication would suggest up to one result being a false positive.

Beyond univariable MR analyses, we attempted to further validate the findings with a multivariable extension of the inverse-variance weighted (IVW) and MR Egger methods, however, in some cases we observed imprecise estimates in line with previous studies which attributed this to the inclusion of highly correlated exposures in the model[41]. To further evaluate if the association signals in the exposure and outcome datasets shared a causal genetic variant, we performed colocalization analyses. Because these analyses were originally developed to find evidence of co-localization between mRNA expression and disease and not for an intermediate trait and disease, the default prior probabilities used in the analysis may not be optimal for these pairs of traits. In addition, the single-causal-variant assumption in genetic co-localization methods may not always be satisfied even when prior conditional analyses are performed, with regions with multiple causal variants potentially yielding false-negative results[42].

The effect directions of the replicated drug targets were compared to results from clinical trials using data from the clinical-trials.gov registry. However, the lack of precision in the annotation of events associated with lipid perturbations (e.g., hyperlipidemia) in this dataset hinders the assignment of reported lipid abnormalities to a particular lipid sub-fraction. Moreover, the proportion of clinical trials with reported results in clinicaltrials.gov is less than 54.2%[43], suggesting that additional drug candidates with lipid effects might have been investigated but were not included in this analysis because of the lack of accessible data. Furthermore, our analysis relied on mapping clinical trial interventions to compounds known to act through binding to the targets of interest, which could potentially miss clinical trials of compounds annotated with fewer synonyms (such as research codes for compounds used by individual trial sponsors). Lastly, we performed a PheWAS spanning over 100 clinical endpoints, 80 of which were derived from UK Biobank.

While this enabled screening for associations with a wide range of diseases, genetic associations derived from diagnostic codes in electronic health record datasets might suffer from limited case numbers and inaccurate case and control definitions, which would reduce the power to detect true associations. To increase the power to detect associations, we included data from publicly available GWAS with the largest sample sizes for such phenotypes.

In summary, we have shown an approach to move from GWAS signals to drug targets and disease indications. We illustrated its potential using genetic association data on lipids and CHD data, but the approach could also be applied in other settings where there are GWAS of diseases and biomarkers thought to be potentially affected by the drug target. For example, with the increasingly available data on inflammatory biomarkers, this approach could be used to evaluate the causal role of anti-inflammatory drug targets, such as IL6R, in CHD, Alzheimer's disease, and major depression, following up on associations described in several studies[44–46], to identify potential new indications for anti-inflammatory agents established in the treatment of autoimmune conditions. Similarly, recent genetic studies on coagulation factor levels[47] can be harnessed to instrument the effect of modulating druggable targets for thrombotic disorders, such as FXI or FXII, which are emerging as potential targets for anticoagulant drugs[48,49].

When used as a screening tool, the approach could help reduce the high failure rate problem in drug discovery by genetically validating targets in the earlier phases of the drug development pipeline.

## Methods

**Data sources**. To determine the causal role and replicate previously reported results on the causal effect of LDL-C, HDL-C, and TG on CHD, we obtained summary-level genetic estimates from the Global Lipids Genetics Consortium (188,577 individuals)[12] and from CardiogramPlusC4D (60,801 cases and 123,504 controls)[13].

Independent replication data were sourced using lipids exposure data from a GWAS meta-analysis of metabolic measures by the University College London–Edinburgh-Bristol (UCLEB) Consortium[15] and Kettunen et al.[16] utilizing NMR spectroscopy measured lipids (joint sample size up to 33,029). Independent CHD data was obtained from a publicly available GWAS of 34,541 cases and 261,984 controls in UK Biobank[17].

Individual-level data from a random subset of 5000 unrelated individuals of European ancestry from UK Biobank was used to generate the LD reference matrices as described in the Instrument selection section.

**Drug target gene selection**. Analyses were conducted using Python v3.7.3. To estimate the causal effect of modulating the level of each lipid sub-fraction via a druggable gene on CHD, variants associated with LDL-C, HDL-C, and/or TG with a $P$ value $\leq 1 \times 10^{-6}$ were selected. Druggable genes overlapping a 50 kbp region around the selected variants were extracted, resulting in 341 associated drug target genes (149 for LDL-C, 180 for HDL-C, and 154 for TG). The set of genes in the druggable genome were identified[18], and identifiers were updated to Ensembl version 95 (GRCh37), used in this analysis. Because we only scanned for genetic associations with the druggable genome, protein-coding genes that were the "true" causal gene but not yet druggable would be missed and the association misassigned. To mitigate this and provide information about potential effects through non-druggable genes, we provide the minimum distance from the variant to the gene, where variants located within a gene were given a distance of 0 bp, a gene distance rank value according to their base-pair distance, and indicated the druggable genes prioritized by GLGC (Supplementary Data 1).

**Instrument selection**. For the genome-wide biomarker MR analyses, a $P$ value threshold of $1 \times 10^{-6}$ was used to select exposure variants associated with LDL-C, HDL-C, and/or TG. For cis- or drug target MR analyses, variants from/within the 341 selected genes ($\pm 50$ kbp) were selected based on a $P$ value $\leq 1 \times 10^{-4}$. In both settings, variants were filtered on a MAF > 0.01 and LD clumped to an $r^2 < 0.4$. These parameters showed the most consistent estimates in a grid-search in the discovery data using the positive control examples: PCSK9, NPC1L1, HMGCR, and CETP (Supplementary Fig. 34). To account for residual correlation between variants in the MR analyses, we applied a generalized least squares framework with an LD reference dataset derived from UK Biobank[50]. LD reference matrices were created by extracting a random subset of 5000 unrelated individuals of European ancestry from UK Biobank. Variants with a MAF < 0.001, and imputation

quality < 0.3 were excluded. To ensure that SNPs with lower MAF have higher confidence, variants were removed if MAF < 0.005 and genotype probability < 0.9; MAF < 0.01 and genotype probability < 0.8; MAF < 0.03 and genotype probability < 0.6.

**MR analysis**. As a validation step, a genome-wide biomarker MR analysis was conducted for each lipid sub-fraction to replicate previous findings using genetic associations across the genome. A model-selection framework was used to decide between competing IVW fixed-effects, IVW random-effects, MR–Egger fixed effects or MR–Egger random-effects models[14]. While IVW models assume an absence of directional horizontal pleiotropy, Egger models allow for possible directional pleiotropy at the cost of power. After removing variants with large heterogeneity ($P$ value < 0.001 for Cochran's $Q$ test) or leverage, we re-applied this model selection framework and used the final model. The influence of parameter selection in the drug target MR performance was explored in a grid-search of several $r^2$ and gene boundaries combinations using the positive control examples PCSK9, NPC1L1, HMGCR, and CETP, where the lipid perturbation is the intended indication. To assess the possibility of false-positive results, we compared the empirical $P$ value distribution of the discovery MR findings against the continuous uniform distribution using the KS goodness-of-fit test (two-sided). Under the null hypothesis of no association, $P$ values follow a continuous uniform distribution between 0 and 1[19].

In addition, we conducted genome-wide biomarker and drug target multivariable MR analyses using genetic associations with the three lipid sub-fractions and CHD risk in a single regression model, to identify likely mediating lipids in the causal pathway of CHD.

Results were presented as mean difference (MD) or OR with 95% confidence interval (95% CI) coded towards the canonical drug target effect direction; i.e., toward lower LDL-C and triglyceride concentration, and higher HDL-C concentration.

**Co-localization analysis**. To estimate the posterior probability of each druggable gene sharing the same causal variant for the exposure lipid and CHD risk[51] we performed colocalization analyses. First, we conducted a stepwise conditional analysis using GCTA-COJO v1.92.4 with genotype data from 5000 individuals randomly selected from UK Biobank[52]. Colocalization analyses were performed using a Python implementation of the Bayesian method "coloc" v3.2-1[20]. The default prior probabilities were used to estimate if an SNP was associated only with the lipid sub-fraction ($p_1 = 10^{-4}$), only with CHD risk ($p_2 = 10^{-4}$), or with both traits ($p_{12} = 10^{-5}$). For each drug target gene, all variants from/within the gene boundaries (±50 kbp) with a MAF > 0.01 were included. A posterior probability above 0.8 was considered sufficient evidence of colocalization based on previous observations[20].

**Drug indications and adverse effects**. To evaluate if the drug target MR and colocalization analyses rediscovered known drug indications, adverse effects, or predicted repurposing opportunities, drug information, and clinical trial data were extracted for the set of 341 druggable targets. Drug target genes were mapped to UniProt identifiers and indications and clinical phase for compounds that bind the target were extracted from the ChEMBL database (version 25)[53]. Drug indications and lipid adverse effects data for licensed drugs were extracted from the British National Formulary (BNF) website (https://bnf.nice.org.uk/) in July 2019.

To further examine the effects of the drugs and clinical candidates that are known to act through binding to the 341 druggable targets, relevant clinical trial data were downloaded from the clinicaltrials.gov registry. Compound names and synonyms were extracted from ChEMBL database (version 25)[53] and used to identify clinical trials with matching interventions. In the case of non-exact matches, the results were inspected manually to ensure that only relevant trial records were used in the analysis. Lipid-related trial outcomes and adverse events were identified by searching the relevant fields within the trial records with the keywords: lipo*, lipid*, ldl*, hdl*, cholest*, and triglyceride*. For adverse events, the search was limited to the trial arm in which the drug of interest was administered (as opposed to placebo or active control used in the study), and only adverse events that affected at least one study participant were included.

**Tissue expression analysis**. To further characterize the genes prioritized by the MR pipeline, their tissue expression was analyzed as follows. First, RNAseq data were downloaded from Human Protein Atlas (HPA)[22], which captures the baseline expression of human genes and proteins across a panel of diverse healthy tissues and organs. For each included gene and tissue, HPA provides a consensus Normalized eXpression value (NX), obtained by normalizing TPM (transcripts per million) values from three independent transcriptomics datasets: GTEx[54], Fantom5[55], and HPA's own RNAseq experiments[56].

The downloaded NX values were then used to investigate if the prioritized target genes were specifically expressed in any of the included tissues. Two commonly used tissue specificity metrics were calculated for each gene: tau and z-score[23]. Tau summarizes the overall tissue distribution of a given gene and ranges from 0 to 1, where 0 indicates ubiquitous expression across all included tissues (house-keeping genes) and 1 indicates narrow expression (highly tissue-specific genes). While tau provides a single summary measure of the tissue specificity, z-scores are calculated for individual tissues separately to quantify how elevated the

gene expression is in a particular tissue compared to others. Here, higher $z$-score values indicate higher tissue specificity. See Kryuchkova-Mostacci et al.[23] for details on the calculation and interpretation of the two metrics.

**Phenome-wide scan of replicated drug target genes**. To provide an overview of potential non-CHD effects of the prioritized drug targets, we performed a phenome-wide scan of 102 disease endpoints. These included genome-wide summary statistics for 80 ICD10 main diagnoses in UK Biobank, with the exclusion of Coronary Artery Disease (ICD10 code: I25), which was explored in detail in the previous sections. The data were released by Neale Lab (1st August 2018, http://www.nealelab.is/uk-biobank/), and downloaded using a Python implementation of MR Base API[57]. The variants in and around the prioritized drug target genes allowing for a boundary region of 50 kbp were extracted, palindromic variants were inferred using the API default MAF threshold of 0.3 and removed[58]. The Ensembl REST Client was used to gather positional information for the variants[59].

Power was further maximized by sourcing data from 23 publicly available GWAS with the largest sample sizes for such phenotypes (Supplementary Table 3 and Supplementary Data 5). All the GWAS clinical endpoints and UK Biobank ICD10 main diagnoses were grouped according to ICD10 chapters.

**Reporting summary**. Further information on research design is available in the Nature Research Reporting Summary linked to this article.

## Data availability

All source GWAS data used throughout the paper are publicly available as specified in the Methods. We used GWAS data for the following outcomes (Supplementary Table 3 and Supplementary Data 5): LDL-C, HDL-C, Triglycerides [http://lipidgenetics.org/#data-downloads-title], Coronary Heart Disease [http://www.cardiogramplusc4d.org/data-downloads/], Rheumatoid arthritis [https://grasp.nhlbi.nih.gov/downloads/ResultsOctober2016/Okada/], Juvenile arthritis [http://ftp.ebi.ac.uk/pub/databases/gwas/summary_statistics/HinksA_23603761_GCST005528/], Ankylosing spondylitis [https://ftp.ebi.ac.uk/pub/databases/gwas/summary_statistics/CortesA_23749187_GCST005529/], Ulcerative colitis [http://ftp.ebi.ac.uk/pub/databases/gwas/summary_statistics/LiuJZ_26192919_GCST003045/], Psoriasis [http://ftp.ebi.ac.uk/pub/databases/gwas/summary_statistics/TsoiLC_23143594_GCST005527/], Crohn disease [http://ftp.ebi.ac.uk/pub/databases/gwas/summary_statistics/LiuJZ_26192919], Stroke [http://www.megastroke.org/index.html], Asthma [https://www.thelancet.com/journals/lanres/article/PIIS2213-2600(18)30389-8/fulltext], Multiple sclerosis (https://www.ncbi.nlm.nih.gov/pmc/articles/PMC3832895/], Gout [http://ftp.ebi.ac.uk/pub/databases/gwas/summary_statistics/TinA_31578528_GCST008970/], Ovarian neoplasms [http://ftp.ebi.ac.uk/pub/databases/gwas/summary_statistics/PhelanCM_28346442_GCST004462/], Parkinson disease [https://drive.google.com/drive/folders/10bGj6HfAXgl-JslpI9ZJIL_JIgZyktxn], Alzheimer disease (https://www.ncbi.nlm.nih.gov/pubmed/30617256], Type 2 diabetes mellitus [https://www.nature.com/articles/s41588-018-0241-6], Myocardial infarction [http://www.cardiogramplusc4d.org/data-downloads/], Heart failure [https://www.nature.com/articles/s41467-019-13690-5], Atrial fibrillation [https://www.nature.com/articles/s41588-018-0171-3], Diabetic nephropathies [http://ftp.ebi.ac.uk/pub/databases/gwas/summary_statistics/vanZuydamNR_29703844_GCST005881], Chronic kidney failure [https://www.ncbi.nlm.nih.gov/pmc/articles/PMC6698888/], Schizophrenia [http://www.med.unc.edu/pgc/files/resultfiles/], Narcolepsy [http://ftp.ebi.ac.uk/pub/databases/gwas/summary_statistics/FaracoJ_23459209_GCST005522/], Atopic dermatitis [http://ftp.ebi.ac.uk/pub/databases/gwas/summary_statistics/PaternosterL_26482879_GCST003184], Biliary liver cirrhosis [http://ftp.ebi.ac.uk/pub/databases/gwas/summary_statistics/CordellHJ_26394269_GCST003129], and 80 ICD10 main diagnoses in UK Biobank released by Neale Lab (1st August 2018, http://www.nealelab.is/uk-biobank/). Data from clinical trials were queried from the clinicaltrials.gov registry. Data on licensed drugs and compounds under development were sourced from the British National Formulary and ChEMBL v25, respectively. The data underlying each figure have been deposited in the UCL Research Data Repository under accession code https://doi.org/10.5522/04/14555715.

## Code availability

The code underlying each figure has been deposited in the UCL Research Data Repository under accession code https://doi.org/10.5522/04/14555715.

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

## Acknowledgements

The authors are grateful to the studies and consortia that provided summary association results and to the participants of the biobanks and research cohorts. This research has been conducted using the UK Biobank Resource under Application Number 12113. UK Biobank was established by the Wellcome Trust medical charity, Medical Research Council, Department of Health, Scottish Government, and the Northwest Regional Development Agency. It has also had funding from the Welsh Assembly Government and the British Heart Foundation. M.G.M. is supported by a BHF Fellowship FS/17/70/33482. A.F.S. is supported by BHF grant PG/18/5033837 and the UCL BHF Research Accelerator AA/18/6/34223. C.F. and A.F.S. received additional support from the National Institute for Health Research University College London Hospitals Biomedical Research Centre. A.D.H. is an NIHR Senior Investigator. We further acknowledge support from the Rosetrees Trust. The UCLEB Consortium is supported by a British

Heart Foundation Program Grant (RG/10/12/28456). M.K. was supported by grants from the Wellcome Trust, UK (221854/Z/20/Z), the UK Medical Research Council (R024227 and S011676), the National Institute on Aging, NIH (R01AG056477 and RF1AG062553), and the Academy of Finland (311492). AH receives support from the British Heart Foundation, the Economic and Social Research Council (ESRC), the Horizon 2020 Framework Program of the European Union, the National Institute on Aging, the National Institute for Health Research University College London Hospitals Biomedical Research Centre, the UK Medical Research Council and works in a unit that receives support from the UK Medical Research Council. A.G. is funded by the Member States of EMBL. P.C. is supported by the Thailand Research Fund (MRG6280088). D.A.L. is supported by a British Heart Foundation Chair (CH/F/20/90003) and British Heart Foundation grant (AA/18/7/34219), is a National Institute of Health Research Senior Investigator (NF-0616-10102) and works in a Unit that receives support from the University of Bristol and UK Medical Research Council (MC_UU_00011/6). This work was funded in part by the UKRI and NIHR through the Multimorbidity Mechanism and Therapeutics Research Collaborative (MR/V033867/1).

## Author contributions

M.G.M., A.D.H., A.F.S., and C.F. contributed to the idea and design of the study. M.G.M. performed the Mendelian randomization and PheWAS analyses. M.G.M. and M.Z. performed the tissue expression and clinical trial analyses. M.G.M., M.Z., A.D.H., A.F.S., and C.F. contributed to the first draft of the paper. M.G.M., M.Z., P.C., F.D., S.C., T.S., J.E., N.C., O.P., G.W., A.W., R.S., M.K., J.F.P., A.D.H., T.R.G., D.A.L., A.G., A.D.H., A.F.S., and C.F. contributed to and approved the final version of the paper.

## Competing interests

A.F.S. has received Servier funding for unrelated work. M.Z. conducted this research as an employee of BenevolentAI. Since completing the work M.Z. is now a full-time employee of GlaxoSmithKline.
