## [Peer Review File · Nature Communications]

REVIEWER COMMENTS

Reviewer #1 (Remarks to the Author):

The paper describes an *in silico* approach for validating drug target, specifically for lipid therapeutics and CHD, using human genomics. The paper is well-written, describing a powerful application of statistical genetics to help address an important clinical problem.

Main comments:

Table 1 could be easily described in the text and Table 2 reports findings from published research/databases. Accordingly, I suggest moving this content to the text and use the Tables to report information about the current data (promoting a couple of the key supplementary tables to the main body of the manuscript).

P7, line 170: Having NPC1L1 and PCKSK9 etc on the prioritized list appears, at face value, to provide positive controls for this approach. However, with 341 genes to work with and multiple outcomes, what are the probabilities that these would be detected by chance (ie false positive rate) and to what extent were other targets that are not on the prioritized list excluded by chance (ie false negative rate)?

In MR analyses of this nature, the genetic instruments rarely (if ever) completely characterize the diversity of the target exposure, owing to the biological complexity of the target. This is certainly the case with almost all lipids used in clinical settings. In some instances, an MR result may support a causal relationship between lipid X and outcome Y, even though it's a small component of lipid X that drives this causal relationship. In other instances, the MR analysis may conclude lack of causal effect, because the non-causal component of the exposure swamps the causal component. Assuming this can go either way, the issue is one of error (rather than bias) and sample size will be the factor that swings the balance. However, working with adequately powered datasets won't alone help dissect the signal, which is important in the context of drug target identification. The authors deal with this problem to some extent through the analysis of lipid subfractions and proteins and in the section "Discriminating independent lipid effects" (p9). However, almost all emphasis is placed in this paper on the statistical significance and relative risk. It would be even more informative to also estimate the fraction of the variance in the classical lipids (and subfractions, for that matter), that are likely to be causal/non-causal in relation to CHD. Even if analyses can't reasonable be performed to address this problem, discussion of the point may be informative for the reader.

An important clinical issue is overlooked in this analysis (and in many drug development pipelines), which is that triglyceride (and certain other lipid) response is heavily influenced by diet, and because most people spend most of their waking hours in the postprandial state, postprandial triglyceride is probably at least as relevant for CHD risk as fasting or random triglycerides, in part because PP triglycerides can be more easily modified through clinical intervention. Unfortunately, postprandial triglyceride response does not appear to be under substantial genetic control (see Berry et al, Nat Med. 2020). Briefly mentioning this point in the Discussion is perhaps warranted.

Minor remark:

Line 67: This comment is very trivial in the context of this paper, but although MR typically uses genetic instruments, one can in principle use other non-time varying instrumental variables (eg sex) in MR studies.

Reviewer #2 (Remarks to the Author):

This manuscript demonstrates the utility of a Drug target Mendelian Randomization approach to prioritize drug targets for their causal relevance on disease endpoints. The specific application in this manuscript is in coronary heart disease and lipid-related therapeutic targets. This approach is very interesting and exciting as it demonstrates a potential strategy to use data from human genetics to prioritize and validate drug targets which could potentially accelerate the drug development process. The work is very well done, scientifically sound, and well written. I think that the community will see tremendous opportunity in this approach for a wide variety of therapeutic and disease areas.

While my enthusiasm for the manuscript is quite high, I have a few areas where the details were not entirely provided or clear. Through these clarifications, I believe that the manuscript will be much improved.

1. In this work, the authors focus on variants that are most likely to associate with gene expression or function in cis. I recognize that even identifying trans effects is a challenge; still I am curious if the authors evaluated any known trans effects for these lipid-related targets? For this work to be most effective, I assume that both cis and trans regulatory effects will be important. Nonetheless, we may not have enough knowledge of the trans effects as of yet. I think it would be worth discussing this issue either more.

2. In the phenome-wide association study, it is not clear how the traits were selected. Did the authors focus on traits correlated with CHD? Or related to the lipid target genes? They looked at 103 disease traits, but it is not clear how this list was generated.

3. In the Results section on independent validation of drug target MR estimates (lines 224-233), how many tests were performed here? It is not clear to me why a p-value threshold of $p < 0.05$ was used. Is there any correction for multiple testing here?

4. Conversely in the PheWAS section, lines 286-297, the authors used a genome-wide significance threshold. It is not clear why this was chosen, when for this study, they did not look at genetic variation genome-wide. Were there other interesting results that did not meet genome-wide significance and thus might be false negatives by using that stringent threshold?

5. One detail that was not clearly described is related to the Data Sources used. Were all of the datasets based on summary statistics only? Or were some of the analyses based on individual-level data? It looks like most of them are summary statistics based, but a few were unclear, especially where the LD data was included. In terms of reproducibility of the study, it would be helpful to a reader who is new to MR, to provide them with a little more detail in the methods.

6. The PheWAS plots look very unusual to me. Why are there curved lines for the p-values for each trait color? How is the X-axis arranged? Typically, the x-axis is by phenotype code, and it would be the same across all of the PheWAS plots. That allows one to look across the plots for genes that show similar patterns. It seems that these are arranged by p-value by trait category? I think it would be best to keep them in the standard PheWAS plot style whereby the x-axis is the same across all the plots.

Reviewer #3 (Remarks to the Author):

dear Authors ,

I have reviewed the article "validation of lipid-related therapeutic targets for coronary heart disease prevention using human genetics "by Gordillo-Maranon and colleagues

As a reviewer I am impressed by this massive undertaking and the multi-pronged approach of Biomarker MR , Drug Target MR , and the Phenome wide scan seems very comprehensive

As far as I can judge the databases used are top of the line , the methodologies very sound and the data are impressive enough to warrant a public report....The manuscript is well written , the results are compact and the Discussion adequate.....

Nevertheless , the devil is in the detail and I would like to illustrate that with a lipid gene that I know very well , CETP , that the details on that specific target and its inhibitors are not always correctly represented....let me make a few points ;

. page 3 ; raising HDL by....was effective in preventing CHD eventsthis opinion is anathema to what the Oxford authors and the rest of the world now believesanacetrapib prevented CHD events through non-HDL lowering and REVEAL fits perfectly on that CHD to non-HDL regression linemoreover , there is now 6.4 year follow up on REVEAL available that reaffirms that CETP-inhibition robustly lowers CHD events through lowering of atherogenic lipoproteins.....please add that ref ...

. page 4 ; the discussion between line 96 and 105 becomes invalid now that we know that CETP-inhibitors lower apoB as well as LDL and that downstream effect reduces CHD....that section needs to be rewritten....

. page 8 ; CETP was associated with lower CHD risk when the effect was instrumented through an elevation of HDL-C.....I already discussed the contrast between LDL and HDL when it comes to explaining how CETPi prevents CHD risk , but in so many of your analyses the HDL angle comes cropping up that there might be an alternative explanation for thatall CETP-inhibitors tested in phase III clinical trials have shown a reduction in HbA1C , Homa-IR and when investigated properly a reduction of new onset diabetes mellitus and in the case of dalcetrapib , a reversal of type II to non-diabetes ..since the only thing that dalcetrapib has an effect on is HDL-C , the anti-diabetic effects of CETPi must occur through raising HDL-C.....possibly through improving cholesterol efflux at the beta cells in the pancreas....so , my next question is ; why does this MR analysis not show a relationship between diabetes and CETP ?

. Last line on page 9 ..there are multiple reports on the fact that low levels of CETP are protective against Alzheimers , especially in carriers of apoE4 alleles....again highlighting the fact that CETP in the periphery (I do not think there is any CETP in the brain) has an effect on neuronal/astrocyte health in the brain.....why is this not seen in this particular analysis ?

. Figure S10 ; the genetic association for LDL-C is strong and tight , supported now by REVEAL , the association for HDL-C is all over the place and I conclude from this figure that CETP-inhibitors and CHD do NOT work through modifying HDL-C.....but I am happily convinced if the authors think otherwise.....the second point in this figure is triglyceridesCETPi does hardly effect TG levelshow can TG modification then have anything to do with CHD prevention ? What is confounding here ?

The purpose of these points is the fact that it is very hard to comment on all prioritized targets in a review of this paper , simply because no one can have all the knowledge on all targets....I would have expected the authors to be better aware of the clinical trials and how they are interpreted , so I expect them to involve clinical colleagues who do know....

Reviewer #4 (Remarks to the Author):

The authors address an interesting and important clinical question.

Background: Drug target Mendelian randomization (MR) studies use DNA sequence variants in or near a gene encoding a drug target, that alter its expression or function, as a tool to anticipate the effect of drug action on the same target.

Study: The authors on top applied MR to prioritize drug targets for their causal relevance for coronary heart disease (CHD).

To corroborate and interpret their results the authors further prioritized the targets using genetic co-localization, protein expression profiles from the Human Protein Atlas and, for targets with a licensed drug or an agent in clinical development, by sourcing data from the British National Formulary and clinicaltrials.gov.

The found that out of the 341 drug targets identified through their association with circulating blood lipids (HDL-C, LDL-C and triglycerides), they were able to robustly prioritize 30 targets that might elicit beneficial treatment effects in the prevention or treatment of CHD. The prioritized list included NPC1L1 and PCSK9, the targets of licensed drugs whose efficacy has been already proven in clinical trials.

The concluded their interesting paper by discussing in depth how this approach could be generalized to other targets, disease biomarkers and clinical end-points to help prioritize and validate targets during the drug development process.

As indicated this reviewer judges the paper as original, innovative and of potential clinical value to the CV-field.

The paper reads well, is well illustrated and well referenced

Especially the different steps of corroborating the evidence (as described above) was appreciated by this reviewer, making it an in my opinion thorough and plausible paper/result.

Needless to say that this type of analyses all have their individual possible shortcomings/limitations. These are well addressed and discussed and put into perspective, and as indicated the total body of results makes sense.

I have no major comments/concerns

J.Wouter Jukema

Reviewer #1

The paper describes an in silico approach for validating drug target, specifically for lipid therapeutics and CHD, using human genomics. The paper is well-written, describing a powerful application of statistical genetics to help address an important clinical problem.

Main comments:

1) Table 1 could be easily described in the text and Table 2 reports findings from published research/databases. Accordingly, I suggest moving this content to the text and use the Tables to report information about the current data (promoting a couple of the key supplementary tables to the main body of the manuscript).

Response: We thank the reviewer for this suggestion and have moved Table 1 to the Supplementary Section (Supplementary Table 1).

However, Table 2 links our novel MR findings to drug indications for the subset of already drugged targets. As such, it provides key information as a positive control, as well as delineating potential repurposing opportunities. Therefore, if the reviewer and editors agree, we prefer to retain the table in the main body as we feel to do so will benefit the reader.

2) P7, line 170: Having NPC1L1 and PCKSK9 etc on the prioritized list appears, at face value, to provide positive controls for this approach. However, with 341 genes to work with and multiple outcomes, what are the probabilities that these would be detected by chance (ie false positive rate) and to what extent were other targets that are not on the prioritized list excluded by chance (ie false negative rate)?

Response: We thank the reviewer for this important comment. We address the multiple testing burden through a number of complementary approaches:

- First, we have now clarified that our main analysis focuses on coronary heart disease (CHD) and we have attempted to address multiplicity for this outcome alone, excluding the outcomes considered later as part of the phenome-wide scan. The outcomes considered in the phenome-wide scan are merely used to provide additional insight on prioritised targets for future follow-up of potential mechanism-based effects on outcomes beyond CHD.
- Second, to assess the potential for false positive results, we note that under the null hypothesis of no association where all results are false positive, P value follow a continuous uniform distribution. Hence, to assess the overall significance of the results, we have compared the empirical P value distribution of the discovery sample against the continuous uniform distribution using Kolmogorov-Smirnov goodness-of-fit test (see Figure below). This comparison shows that our observed P value distribution differs considerably from the uniform distribution. As such, we made the following amendments to the manuscript:

Page 7:

“To assess the potential for false positive results, the distribution of the exposure-specific P values was tested against the uniform distribution expected under the null hypothesis¹⁹. The Kolmogorov-Smirnov (KS) goodness-of-fit test was not consistent with the hypothesis that the observed findings could be readily explained by multiple testing (Supplementary Fig. 2).”

Page 16:

“To assess the potential for false positive results, we tested the distribution of the exposure-specific P values against the uniform distribution expected under the null hypothesis¹⁹. The Kolmogorov-Smirnov (KS) goodness-of-fit test indicated that the number of extreme P values obtained would be highly unlikely under the null hypothesis, suggesting that they are unlikely to represent false positives”

Page 21:

“To assess the possibility of false positive results, we compared the empirical P value distribution of the discovery MR findings against the continuous uniform distribution using the Kolmogorov-Smirnov goodness-of-fit test. Under the null hypothesis of no association, P values follow a continuous uniform distribution between 0 and 1¹⁹.”

And the following figure has now been added to the Supplementary Material:

“Supplementary Figure 2. Density distribution of the P values in the discovery analysis by exposure. Kolmogorov-Smirnov (KS) goodness-of-fit test against the continuous uniform distribution of P values (black dashed line) expected under the null-hypothesis of no association between any of the targets and coronary heart disease, when the effect is instrument via LDL-C, HDL-C and TG effects”

- Third, we validated the CHD focused drug target MR analyses using two sources of independent data: i) lipid measurements from GLGC against CHD estimates from CardiogramPlusC4D and ii) NMR spectroscopy measured lipids from UCLEB against CHD estimates from UK Biobank; referred to as discovery and replication, respectively. For the 98 drug targets with data in both the discovery and replication samples, we constructed a prioritized list with the targets that showed directionally concordant results with CHD and were significant (at an $\alpha = 0.05$) in both analyses. By drawing inference on the replicated data, the multiple testing burden was considerably reduced, as the probability of getting two significant results under the null hypothesis becomes $0.05^2=0.0025$. If we now multiply this new threshold by the 98 drug targets retained after replication, we would, on average, expect $98 \times 0.05^2 = 0.245$ false positive results. Hence, our response to the reviewer’s concern is that we would expect up to one false positive result. The following was text was added to the manuscript to address this issue:

Page 9:

“To help verify the MR findings and reduce the multiple testing burden...”

Page 16:

“By drawing inference on replicated data, the multiple testing burden was considerably reduced ($0.05^2=0.0025$), which when applied to 98 drug targets retained after replication would suggest up to one result being a false positive.”

3) In MR analyses of this nature, the genetic instruments rarely (if ever) completely characterize the diversity of the target exposure, owing to the biological complexity of the target. This is certainly the case with almost all lipids used in clinical settings. In some instances, an MR result may support a causal relationship between lipid X and outcome Y, even though it's a small component of lipid X that drives this causal relationship. In other instances, the MR analysis may conclude lack of causal effect, because the non-causal component of the exposure swamps the causal component. Assuming this can go either way, the issue is one of error (rather than bias) and sample size will be the factor that swings the balance. However, working with adequately powered datasets won't alone help dissect the signal, which is important in the context of drug target identification. The authors deal with this problem to some extent through the analysis of lipid subfractions and proteins and in the section “Discriminating independent lipid effects” (p9). However, almost all emphasis is placed in this paper on the statistical significance and relative risk. It would be even more informative to also estimate the fraction of the variance in the classical lipids (and subfractions, for that matter), that are likely to be causal/non-causal in relation to CHD. Even if analyses can't reasonable be performed to address this problem, discussion of the point may be informative for the reader.

Response: We thank the reviewer for this very relevant point on inference.

First, we agree fully that a non-significant result should not be interpreted as a proof of absence of an effect. Therefore, throughout the manuscript, we flagged significant associations without emphasizing on non-significant results to avoid this being incorrectly interpreted as making an argument for the absence of an effect. In addition, we have further improved to manuscript by removing any potentially confusing statements on “null effects”.

We have added the following to the main manuscript to further emphasize this:

Page 13:

“In general, results that do not meet the significance threshold should not be over-interpreted as proof of absence of effect²⁹. This may be exacerbated here by potential weak instrument bias, which would be expected to attenuate results towards the no-effect direction”

Second, taking forward the reviewer's point on the 'complexity of the target', we notice that our original submission did not sufficiently clarify that our inferential target is the protein drug target and not the lipid fraction used to weight the effect of perturbing the drug target. Thus, the univariable, *cis* genetic associations weighted by LDL-C, HDL-C, and TG are used as proxies of the protein level or function and do not to provide direct evidence on the mediator. Indeed, such protein effects might act through (one, or multiple) lipid pathways, or completely sidestep these pathways entirely. However, this is a secondary issue when the question framed is on the validity of the drug target, since any drug targeting the same protein is anticipated to share the same mediating pathways, whatever these are. Recently, we have developed a formal mathematical derivation for this argument in an earlier *Nature Communications* paper (Schmidt *et al.*, 2020). To illustrate the argument presented in our 2020 paper, we have included the following diagram, where a genetic variant **G**, the encoded protein **P**, and the downstream biomarker **X** (e.g., a lipid fraction) are shown, any or all of which may have an effect on disease **D**.

Fig. 1 Directed acyclic graphs of potential Mendelian randomisation pathways. Nodes are presented in bold face, with **G** representing a genetic variant, **P** a protein drug target, **X** a biomarker, **D** the outcome, and **U** (potentially unmeasured) common causes of both **P**, **X**, **D**. Labelled paths represent the (causal) effects between nodes.

To see why a biomarker weighted drug target MR (where the genetic effect on X is indicated by $\tilde{\delta}\mu$) does not provide evidence for biomarker X causing disease D , let alone mediation of the $P \rightarrow D$ pathway, we can consider the case where the biomarker itself does not cause disease; that is when $\theta = 0$.

In this case the only remaining pathway from P to disease is ϕ_P , and the “biomarker weighted” drug target MR (ω_{bw}), in the absence of horizontal pleiotropy (that is, when $\phi_G = 0$), is as follows:

$$\begin{aligned}\omega_{bw} &= \frac{\tilde{\delta}(\phi_P + \mu\theta)}{\tilde{\delta}\mu} \\ &= \frac{\phi_P + \mu\theta}{\mu} \quad (\phi_P \text{ cancels out}) \\ &= \frac{1}{\mu} \cdot \phi_P \quad (\text{because } \theta = 0)\end{aligned}$$

Here we note that the final term involves μ (the protein effect on the biomarker), and ϕ_P (the protein effect on disease), but not θ (the biomarker effect on disease, which is null). These derivations show that a “biomarker” weighted drug target MR analysis does not provide evidence on the biomarker effect on disease (which can be null). The biomarker is simply providing a proxy for the level of the effect of G on P and can be useful when P is unmeasured. As such it cannot provide evidence on which biomarker mediates the effect of P on D . As we explain below such mediation can however be explored using multivariable MR (MVMR).

Therefore, the following points were added to further clarify the drug target MR framework and the inferences derived from this analysis:

Page 3:

“Whereas genome-wide biomarker MR helps infer the causal relevance of a biomarker for a disease, a drug target MR helps infer whether, and in certain cases in what direction, a drug that acts on the encoded protein, whether an antagonist, agonist, activator or inhibitor, will alter disease risk (Supplementary Table 1).”

Page 4:

“Importantly, as discussed in detail elsewhere³, drug target MR analyses which use genetic associations with “biomarkers” downstream to the protein, such as HDL-C, use this effect as a proxy for protein concentration or activity (where this has not been measured directly), and do not provide evidence on whether the biomarker used for the weighting itself mediates disease. Rather, they inform on the validity of the drug target for a disease, regardless of the mediating pathway.

Taken together, these observations suggest other similarly effective, but as yet unexploited drug targets might exist for the prevention or treatment of CHD that could be identified through their association with blood lipids even though such analyses need not presume that the effect on CHD is mediated through these lipids.”

Additionally, we have meticulously scrutinised the text to ensure that we refrain from making inference on the mediation pathway when discussing the univariable MR results. Since the questions on potential lipid mediation are very relevant, we have further highlighted that multivariable MR (MVMR) was applied to provide information on the potential lipid mediation pathway the protein may affect CHD through. The following was added to the main text:

Page 5:

“Because of interest in this area, though not the focus of the work, we did also evaluate potential mediators of these effects using multivariable MR (MVMR). Finally, we discuss how this approach might be generalized to other drug targets and clinical endpoints and used to translate findings from GWAS into new drug development.”

Page 6:

“Here we used genetic associations with LDL-C, HDL-C, and TG as a proxy for drug target effects on CHD, which does not provide direct evidence on whether the drug target itself affects CHD through the leveraged lipid weight; this mediation question is subsequently explored using multivariable MR.”

Page 10:

“After considering each lipid sub-fraction as a single measure on disease risk in the univariable drug target MR analyses, we performed a multivariable drug target MR (MVMR) analysis including LDL-C, HDL-C and TG in a single model to account for potential pleiotropic effects of target perturbation via the other lipid sub-fractions and, in contrast to the previous univariable drug target MR, attempt to directly identify the potential lipid mediating pathway.”

Page 13:

“Nevertheless, multivariable MR provides insight on the potential relevance of lipid pathways in mediating the effects of drug target perturbation.”

Page 18:

“We illustrated its potential using genetic association data on lipids and CHD data, but the approach could also be applied in other settings where there are GWAS of diseases and biomarkers thought to be potentially affected by the drug target.”

Page 21:

“Additionally, we conducted biomarker and drug target multivariable MR analyses using genetic associations with the three lipid sub-fractions and CHD risk in a single regression model, to identify likely mediating lipids in the causal pathway of CHD.”

And the following removed from page 12 to avoid confusion:

“Importantly, these effects were observed not only for genes associated with LDL-C, but also TG or HDL-C”

Thirdly, we thank the reviewer for allowing us to provide further explanation on the estimation of the explained variance. Traditionally, the explained variance (R-squared), the F-statistic and/or the P-value of the genetic effect with the exposure have been used as a measure of instrument strength. These metrics are however proportional to each other as detailed in Power and instrument strength requirements for Mendelian randomization studies using multiple genetic variants (Pierce *et al.*, 2011). <https://www.ncbi.nlm.nih.gov/pmc/articles/PMC3147064/pdf/dyq151.pdf>. In our study, we select instruments with an F-statistic of at least 15; where 1/F-statistic is the expected relative bias due to potential weak instrument bias. Given the proportionality between these measures, additional reporting of the R-squared does not provide further information. Therefore, we prefer to report the F-statistic due to its interpretation as a marker of relative bias.

4) An important clinical issue is overlooked in this analysis (and in many drug development pipelines), which is that triglyceride (and certain other lipid) response is heavily influenced by diet, and because most people spend most of their waking hours in the postprandial state, postprandial triglyceride is probably at least as relevant for CHD risk as fasting or random triglycerides, in part because PP triglycerides can be more easily modified through clinical intervention. Unfortunately, postprandial triglyceride response does not appear to be under substantial genetic control (see Berry *et al.*, *Nat Med.* 2020). Briefly mentioning this point in the Discussion is perhaps warranted.

Response: We thank the reviewer for this suggestion. Apologies for any confusion caused on the inferential target but we would like to reiterate that the lipid fractions including triglycerides were used as proxies to weight the effect of genetic variation on the protein target and not to determine the causal relevance or not of triglycerides in CHD for which genome-wide biomarker MR would be more appropriate as mentioned in the Introduction. We used genetic effects estimates for triglycerides either from the Global Lipid Genetics Consortium or from the UCLEB consortium using standard methods and NMR spectroscopy respectively. Because some but not all of the studies contributing to these consortia measured blood lipids on a fasting sample, we are unable to conduct separate analyses based on genetic effects in the fasting and non-fasting state. We have added the following on page 15:

“Lastly, because some but not all of the studies contributing to these consortia measured blood lipids on a fasting sample, we are unable to conduct separate analyses based on genetic effects in the fasting and non-fasting state.”

Minor remark:

5) Line 67: This comment is very trivial in the context of this paper, but although MR typically uses genetic instruments, one can in principle use other non-time varying instrumental variables (eg sex) in MR studies.

Response: We fully agree with the reviewer that MR can be seen as a specific application of instrumental variable (IV) analyses. IV analyses, while particularly popular in econometrics and social science, have seen some uptake in pharmaco-epidemiology using instruments such as prescribing preference of the treating physician. Sex is indeed a very interesting potential instrument, which of course, is closely linked to genetics – although not often used in practice. We have made no amendments to the paper in relation to this point.

Reviewer #2

This manuscript demonstrates the utility of a Drug target Mendelian Randomization approach to prioritize drug targets for their causal relevance on disease endpoints. The specific application in this manuscript is in coronary heart disease and lipid-related therapeutic targets. This approach is very interesting and exciting as it demonstrates a potential strategy to use data from human genetics to prioritize and validate drug targets which could potentially accelerate the drug development process. The work is very well done, scientifically sound, and well written. I think that the community will see tremendous opportunity in this approach for a wide variety of therapeutic and disease areas.

While my enthusiasm for the manuscript is quite high, I have a few areas where the details were not entirely provided or clear. Through these clarifications, I believe that the manuscript will be much improved.

6) 1. In this work, the authors focus on variants that are most likely to associate with gene expression or function in *cis*. I recognize that even identifying *trans* effects is a challenge; still I am curious if the authors evaluated any known *trans* effects for these lipid-related targets? For this work to be most effective, I assume that both *cis* and *trans* regulatory effects will be important. Nonetheless, we may not have enough knowledge of the *trans* effects as of yet. I think it would be worth discussing this issue either more.

Response: We thank the reviewer for allowing us to provide further explanation on why we did not include *trans* variants in the study. The reviewer is correct that when the exposure of interest is the level or function of a protein, genetic instruments used in a drug target MR analysis located in or near the protein of interest (i.e. acting in *cis*) are one class of potential instruments. The other category of instruments that might be used in such an MR analysis are those located outside the encoding gene (i.e. acting in *trans*).

However, *trans* instruments should only be considered for a protein exposure and used to weight the effect of the instrument when the protein itself has been measured directly. Here, we use biomarker effects, distal to the protein of interest, to weight the effect of the genetic instrument. This is legitimate in *cis*-MR analysis in which the genetic instruments used act through the protein of interest. However, unless the protein itself is measured directly, there can be no guarantee in a biomarker weighted *trans*-MR analysis that the *trans* instruments used are acting through the protein of interest. Indeed, the safest assumption in that scenario would be that an instrument of this type is actually acting in *cis* for the protein encoded by the gene in which it resides, and not through the protein of interest.

On the other hand, a *trans* instrument might be considered if the protein of interest is measured directly, since this measure can be used directly to weight the effect of either a *cis*- or *trans*-instrument. However, as we have argued elsewhere (Schmidt *et al.*, 2020 and Swerdlow *et al.*, 2016), even if the protein of interest has been measured directly, the use of a *trans* instrument runs a greater risk of introducing horizontal pleiotropy since the effect of a *trans* instrument on a disease endpoint could be mediated by a pathway that is independent of the protein of interest (i.e. through the protein encoded by the gene in which the *trans* instrument resides). For these reasons, we have focused exclusively on *cis*-instruments in this analysis.

To further clarify the rationale here, we make the following amendments to the manuscript:

Page 16:

“We used cis-MR to evaluate the relevance of each drug target to CHD, which is less prone to violation of the horizontal pleiotropy assumption than MR analyses with trans instruments³, which also

require direct measurement of the protein of interest. However cis-MR analysis also requires some decisions to be made regarding instrument selection”.

7) 2. In the phenome-wide association study, it is not clear how the traits were selected. Did the authors focus on traits correlated with CHD? Or related to the lipid target genes? They looked at 103 disease traits, but it is not clear how this list was generated.

Response: We thank the reviewer and have clarified our disease selection strategy:

Page 12:

“The 102 traits were agnostically selected and represent the entire spectrum of clinical disease available from the Neale Lab UK Biobank release, with the exclusion of Coronary Artery Disease (ICD10 code: I25).”

Page 23:

“To provide an overview of potential non-CHD effects of the prioritized drug targets, we performed a phenome-wide scan of 102 disease endpoints. These included genome-wide summary statistics for 80 ICD10 main diagnoses in UK Biobank, with the exclusion of Coronary Artery Disease (ICD10 code: I25), which was explored in detailed in the previous sections. The data were released by Neale Lab (1st August 2018, <http://www.nealelab.is/uk-biobank/>), and downloaded using a Python implementation of MR Base API⁵⁸. The variants in-and-around the prioritized drug target genes allowing for a boundary region of 50kbp were extracted, palindromic variants were inferred using the API default MAF threshold of 0.3 and removed⁵⁹. The Ensembl REST Client was used to gather positional information for the variants⁶⁰.

Power was further maximized by sourcing data from 23 phenotypically specific GWAS (table S6). All the GWAS clinical endpoints and UK Biobank ICD10 main diagnoses were grouped according to ICD10 chapters.”

8) 3. In the Results section on independent validation of drug target MR estimates (lines 224-233), how many tests were performed here? It is not clear to me why a p-value threshold of $p < 0.05$ was used. Is there any correction for multiple testing here?

Response: We thank the reviewer for this important comment. Please see our response to comment 2 from Reviewer #1, where we explain how multiple testing is addressed. We therefore explain the complementary ways to address multiple testing by making the following amendments:

Page 16:

“We addressed multiple testing in the MR analyses in a number of complementary ways”

Page 7:

“To assess the potential for false positive results, the distribution of the exposure-specific P values was tested against the uniform distribution expected under the null hypothesis¹⁹. The Kolmogorov-Smirnov (KS) goodness-of-fit test was not consistent with the hypothesis that the observed findings could be readily explained by multiple testing (Supplementary Fig. 2).”

Page 16:

“To assess the potential for false positive results, we tested the distribution of the exposure-specific *P* values against the uniform distribution expected under the null hypothesis¹⁹. The Kolmogorov-Smirnov (KS) goodness-of-fit test indicated that the number of extreme *P* values obtained would be highly unlikely under the null hypothesis, suggesting that they are unlikely to represent false positives”

Page 21:

“To assess the possibility of false positive results, we compared the empirical *P* value distribution of the discovery MR findings against the continuous uniform distribution using the Kolmogorov-Smirnov goodness-of-fit test. Under the null hypothesis of no association, *P* values follow a continuous uniform distribution between 0 and 1¹⁹.”

And the following figure was added to the Supplementary Material:

“Supplementary Figure 2. Density distribution of the *P* values in the discovery analysis by exposure. Kolmogorov-Smirnov (KS) goodness-of-fit test against the continuous uniform distribution of *P* values (black dashed line) expected under the null-hypothesis of no association between any of the targets and coronary heart disease, when the effect is instrument via LDL-C, HDL-C and TG effects”

Page 16:

“By drawing inference on replicated data, the multiple testing burden was considerably reduced ($0.05^2=0.0025$), which when applied to 98 drug targets retained after replication would suggest up to one result being a false positive”

9) 4. Conversely in the PheWAS section, lines 286-297, the authors used a genome-wide significance threshold. It is not clear why this was chosen, when for this study, they did not look at genetic variation genome-wide. Were there other interesting results that did not meet genome-wide significance and thus might be false negatives by using that stringent threshold?

Response: We thank the reviewer for the comment and agree that there may be interesting results that did not meet genome-wide significance. However, since there is not a wide consensus on the significance threshold used for phenome-wide association studies and the purpose of including a phenome-wide association scan in this manuscript was exploratory in nature, we decided to report only associations that met genome-wide significance in the Results section. To facilitate the identification of traits in the PheWAS plot, including those that did not meet genome-wide significance, we have added a table in the Supplementary Section (Supplementary Table 8) with the list of traits surveyed in the PheWAS, ordered by their position in the x-axis of the PheWAS plot.

10) 5. One detail that was not clearly described is related to the Data Sources used. Were all of the datasets based on summary statistics only? Or were some of the analyses based on individual-level data? It looks like most of them are summary statistics based, but a few were unclear, especially where the LD data was included. In terms of reproducibility of the study, it would be helpful to a reader who is new to MR, to provide them with a little more detail in the methods.

Response: We thank the reviewer for this comment, and clarified the accessibility of data sources as suggested:

Page 19:

“To determine the causal role and replicate previously reported results on the causal effect of LDL-C, HDL-C and TG on CHD, we obtained summary-level genetic estimates from the Global Lipids Genetics Consortium (188,577 individuals)¹³ and from CardiogramPlusC4D (60,801 cases and 123,504 controls)¹³.

Independent replication data were sourced using lipids exposure data from a GWAS meta-analysis of metabolic measures by the University College London–Edinburgh-Bristol (UCLEB) Consortium⁵⁰ and Kettunen et al¹⁶ utilizing NMR spectroscopy measured lipids (joint sample size up to 33,029). Independent CHD data was obtained from a publicly available GWAS of 34,541 cases and 261,984 controls in UK Biobank¹⁷.

Individual-level data from a random subset of 5,000 unrelated individuals of European ancestry from UK Biobank was used to generate the LD reference matrices as described in the Instrument selection section.”

11) 6. The PheWAS plots look very unusual to me. Why are there curved lines for the p-values for each trait color? How is the X-axis arranged? Typically, the x-axis is by phenotype code, and it would be the same across all of the PheWAS plots. That allows one to look across the plots for genes that show similar patterns. It seems that these are arranged by p-value by trait category? I think it would be best to keep them in the standard PheWAS plot style whereby the x-axis is the same across all the plots.

Response: We thank the reviewer for this suggestion. Accordingly, we have updated all the PheWAS plots to the standard style, where the x-axis represents the same phenotype.

Reviewer #3

dear Authors ,

I have reviewed the article "validation of lipid-related therapeutic targets for coronary heart disease prevention using human genetics "by Gordillo-Maranon and colleagues

As a reviewer I am impressed by this massive undertaking and the multi-pronged approach of Biomarker MR , Drug Target MR , and the Phenome wide scan seems very comprehensive

As far as I can judge the databases used are top of the line , the methodologies very sound and the data are impressive enough to warrant a public report....The manuscript is well written , the results are compact and the Discussion adequate.....

Nevertheless , the devil is in the detail and I would like to illustrate that with a lipid gene that I know very well , CETP , that the details on that specific target and its inhibitors are not always correctly represented....let me make a few points ;

12) page 3 ; raising HDL by....was effective in preventing CHD eventsthis opinion is anathema to what the Oxford authors and the rest of the world now believesanacetrapib prevented CHD events through non-HDL lowering and REVEAL fits perfectly on that CHD to non-HDL regression linemoreover , there is now 6.4 year follow up on REVEAL available that reaffirms that CETP-inhibition robustly lowers CHD events through lowering of atherogenic lipoproteins.....please add that ref ...

Response: We thank the reviewer for raising these points and for allowing us to improve this section, as we agree that the original phrasing lacks nuance and we did not seek to imply that HDL-C elevation necessarily mediates this effect of CETP inhibition on CHD. The original submission did not sufficiently clarify that our inferential target is the (protein) drug target and thus, the LDL-C, HDL-C, and TG *cis* genetic associations are used as downstream proxies of the protein function. As such, the univariable drug target MR analyses do not provide direct evidence on the mediating lipid or lipids, or other non-lipid mediators, and instead provide direct evidence on the effect of drug target perturbation on disease which is the directly relevant question for drug target prioritisation. As noted in the response to Reviewer #1, we have formally derived this approach in the *Nature Communications* paper by Schmidt *et al.*, 2020. Please see our response to comment 2 from Reviewer #1 for a detailed explanation of the framework, which shows that a “biomarker” weighted drug target MR analysis does not provide evidence on the biomarker effect on disease (which can be null). The biomarker weighted *cis*-MR is used to proxy protein concentration or activity *when this has not been measured directly*. As such, it cannot provide evidence on which biomarker *mediates* the effect on any specific outcomes.

To reflect these points, we have amended the text on page 3 as follows:

“A well-established role of Mendelian randomization (MR) analysis is to use such genetic variants (mostly identified from GWAS) as instrumental variables to identify which disease biomarkers (e.g., blood lipids such as low- and high-density lipoprotein cholesterol and triglycerides) may be causally related to disease endpoints (e.g. coronary heart disease; CHD)^{1,2}. We and others have shown that variants in a gene encoding a specific drug target, that alter its expression or function, can be used as a tool to anticipate the effect of drug action on the same target. We have referred to this application of Mendelian randomization as ‘drug target MR’³. In contrast to a genome-wide biomarker MR, where the variants comprising the genetic instrument are selected from across the genome, in a drug target MR analysis, variants are selected from the gene of interest or neighbouring genomic region because these variants are most likely to associate with the expression or function of the encoded protein (acting in cis). Whereas genome-wide biomarker MR helps infer the causal relevance of a biomarker

for a disease, a drug target MR helps infer whether and, in certain cases in what direction, a drug that acts on the encoded protein, whether an antagonist, agonist, activator or inhibitor, will alter disease risk (Supplementary Table 1).

Genome-wide biomarker MR studies have validated the causal role of elevated low-density lipoprotein cholesterol (LDL-C) on coronary heart disease risk, supporting the findings from randomized controlled trials of different LDL-C lowering drug classes⁴⁻⁹. However, such studies have been equivocal on the role of high-density lipoprotein cholesterol (HDL-C) and triglycerides (TG) in CHD^{4,5}. Clinical trials of these lipid fractions have also been seemingly contradictory. For example, using niacin to raise HDL-C did not reduce CHD risk¹⁰, but inhibiting cholesteryl ester transfer protein (CETP) with anacetrapib, which also raises HDL-C, was effective in preventing CHD events¹¹. However, a drug target MR of CETP on CHD, using variants in the CETP gene weighted by their effect on HDL-C, indicates protection from disease (odds ratio: 0.87; 95%CI: 0.84, 0.90)³. The finding is consistent with the effect of allocation to the CETP-inhibitor anacetrapib in a placebo-controlled trial (0.93; 95%CI: 0.86, 0.99) and is compatible with the view that targeting CETP is an effective therapeutic approach to prevent CHD (Fig. 1)¹¹. Importantly, as discussed in detail elsewhere³, drug target MR analyses which use genetic associations with “biomarkers” downstream to the protein, such as HDL-C, use this effect as a proxy for protein concentration or activity (where this has not been measured directly), and do not provide evidence on whether the biomarker used for the weighting itself mediates disease. Rather, they inform on the validity of the drug target for a disease, regardless of the mediating pathway.

Taken together, these observations suggest other similarly effective as yet unexploited drug targets might exist for the prevention or treatment of CHD that could be identified through their association with blood lipids even though such analyses need not presume that the effect on CHD is mediated through these lipids.“

13) page 4 ; the discussion between line 96 and 105 becomes invalid now that we know that CETP-inhibitors lower apoB as well as LDL and that downstream effect reduces CHD....that section needs to be rewritten..

Response: We thank the reviewer for this comment. As explained in the previous comment, the inferential target is the (protein) drug target and thus, the LDL-C, HDL-C, and TG *cis* genetic associations are used as downstream proxies of the protein function. We would like to note that we have evaluated in an independent manuscript the effect of instrumenting CETP through circulating protein levels compared to the instrumentation through lipid subfractions (including Apo-B) using univariable and multivariable MR. In such analysis, which is under review by *Nature Communications* (Manuscript reference number: NCOMMS-20-37421), we concluded that “Supplanting LDL-C by genetic associations with Apo-B, we observed suggestive, but insufficiently precise, evidence of Apo-B mediating the CETP effect on CHD OR 0.60 per SD decrease in Apo-B: 95%CI 0.34; 1.03”.

14) page 8 ; CETP was associated with lower CHD risk when the effect was instrumented through an elevation of HDL-C.....I already discussed the contrast between LDL and HDL when it comes to explaining how CETPi prevents CHD risk , but in so many of your analyses the HDL angle comes cropping up that there might be an alternative explanation for thatall CETP-inhibitors tested in phase III clinical trials have shown a reduction in HbA1C , Homa-IR and when investigated properly a reduction of new onset diabetes mellitus and in the case of dalcetrapib , a reversal of type II to non-diabetes ..since the only thing that dalcetrapib has an effect on is HDL-C , the anti-diabetic effects of CETPi must occur through raising HDL-C.....possibly through improving cholesterol efflux at the beta

cells in the pancreas....so , my next question is ; why does this MR analysis not show a relationship between diabetes and CETP ?

Response: We thank the reviewer for this comment. However, we would like to note that the MR analyses conducted in this manuscript did not include diabetes as an outcome and they just evaluated the drug targets in the context of coronary heart disease. Therefore, the MR analyses presented in this manuscript cannot be used to conclude or exclude a causal relationship between diabetes and any of the drug targets explored, including CETP. Moreover, we would like to note that we have evaluated the association of CETP and multiple clinical endpoints, including diabetes, in an independent manuscript which is currently also under revision for *Nature Communications* (Manuscript reference number: NCOMMS-20-37421). In that manuscript, we compared the effect of *cis*-MR for CETP instrumented through direct measurement of circulating CETP concentration, with that instrumented through lipid subfractions (and Apo-B), using univariable and multivariable MR. We concluded that “MVMR models for CETP indicate its T2DM protective effect likely acts through HDL-C, independent of either LDL-C or Apo-B.”

15) . Last line on page 9 ..there are multiple reports on the fact that low levels of CETP are protective against Alzheimers , especially in carriers of apoE4 alleles....again highlighting the fact that CETP in the periphery (I do not think there is any CETP in the brain) has an effect on neuronal/astrocyte health in the brain.....why is this not seen in this particular analysis ?

Response: We thank the reviewer for this comment. We would like to note that we have evaluated the association of CETP with multiple clinical endpoints, including Alzheimer’s disease, in the previously mentioned manuscript, which is under revision for *Nature Communications* (Manuscript reference number: NCOMMS-20-37421). In that manuscript, we provide an in-depth analysis of CETP as a drug target and discuss the association with Alzheimer’s disease in detail. For the current manuscript, we have simply ensured that type 2 diabetes and Alzheimer’s disease are included in the PheWAS analysis. We illustrate in the figure below that type 2 diabetes and Alzheimer’s disease were indeed covered by the PheWAS of CETP, and if they were not reported in Supplementary Figure 10 is because they did not meet the pre-defined significance threshold. Inspired by suggestions made by Reviewer #2, we have added a Supplementary Table 8 to help the reader identify the diseases alongside the X-axis of the PheWAS plot, particularly those that were not reported because of the significance threshold.

16) Figure S10 ; the genetic association for LDL-C is strong and tight , supported now by REVEAL , the association for HDL-C is all over the place and I conclude from this figure that CETP-inhibitors and CHD do NOT work through modifying HDL-C.....but I am happily convinced if the authors think otherwise.....the second point in this figure is triglyceridesCETPi does hardly effect TG levelshow can TG modification then have anything to do with CHD prevention ? What is confounding here ?

Response: We thank the reviewer for this comment. The purpose of the top and middle left panels of Supplementary Figure 10 is simply to show the different patterns of locus-specific and genome-wide associations, and thus, these panels should not be used to infer either drug target or mediation effects. As further clarified in the response to comment 12, our inferential target is the (protein) drug target and thus, the LDL-C, HDL-C, and TG *cis* genetic associations are used as downstream proxies of the protein function. Therefore, that “biomarker” weighted drug target MR analysis does not provide evidence on the biomarker effect on disease (which can be null). The biomarker weighted *cis*-MR is used to proxy protein concentration or activity *when this has not been measured directly*. These clarifications were also included in the main manuscript as indicated in the response to comment 12.

17) The purpose of these points is the fact that it is very hard to comment on all prioritized targets in a review of this paper , simply because no one can have all the knowledge on all targets....I would have expected the authors to be better aware of the clinical trials and how they are interpreted , so I expect them to involve clinical colleagues who do know....

Response: Of the 24 authors, six are clinically qualified, two are clinically active, four have expertise in the evaluation of clinical trials data, and one is a specialist in Clinical Pharmacology and Therapeutics. Thus we believe the expertise the reviewer refers to is adequately covered by the existing authors.

Reviewer #4

The authors address an interesting and important clinical question.

Background: Drug target Mendelian randomization (MR) studies use DNA sequence variants in or near a gene encoding a drug target, that alter its expression or function, as a tool to anticipate the effect of drug action on the same target.

Study: The authors on top applied MR to prioritize drug targets for their causal relevance for coronary heart disease (CHD).

To corroborate and interpret their results the authors further prioritized the targets using genetic co-localization, protein expression profiles from the Human Protein Atlas and, for targets with a licensed drug or an agent in clinical development, by sourcing data from the British National Formulary and clinicaltrials.gov.

The found that out of the 341 drug targets identified through their association with circulating blood lipids (HDL-C, LDL-C and triglycerides), they were able to robustly prioritize 30 targets that might elicit beneficial treatment effects in the prevention or treatment of CHD. The prioritized list included NPC1L1 and PCSK9, the targets of licensed drugs whose efficacy has been already proven in clinical trials.

The concluded their interesting paper by discussing in depth how this approach could be generalized to other targets, disease biomarkers and clinical end-points to help prioritize and validate targets during the drug development process.

As indicated this reviewer judges the paper as original, innovative and of potential clinical value to the CV-field.

The paper reads well, is well illustrated and well referenced

Especially the different steps of corroborating the evidence (as described above) was appreciated by this reviewer, making it an in my opinion thorough and plausible paper/result.

Needless to say that this type of analyses all have their individual possible shortcomings/limitations. These are well addressed and discussed and put into perspective, and as indicated the total body of results makes sense.

I have no major comments/concerns

J.Wouter Jukema

Response: We are grateful for the supportive comments by the reviewer.

REVIEWERS' COMMENTS

Reviewer #1 (Remarks to the Author):

The authors have adequately addressed most of my comments. A couple of issues persist:

Point 1. The suggestion to keep Table 2 in the main body of the paper is fine with me.

Point 2. In response to a concern I expressed about false discovery (and multiple testing), the authors state that "The outcomes considered in the phenome-wide scan are merely used to provide additional insight on prioritised targets for future follow-up of potential mechanism-based effects on outcomes beyond CHD". I have to say that this explanation doesn't fly. This is somewhat philosophical, but probabilistic tendencies aren't affected by whether one considers a test primary or secondary to ones interests. Accordingly, the burden of testing does include all tests performed, and this needs to be taken into account in any correction procedure. I do however agree that the replication analysis is reassuring.

Point 3. Thank you for this thoughtful and interesting response!

Reviewer #2 (Remarks to the Author):

The authors did a thorough and thoughtful job responding to the previous round of review. I have no further comments. It is a very good paper.

Reviewer #3 (Remarks to the Author):

I am very happy with the responses by the authors

Reviewer #1 (Remarks to the Author)

The authors have adequately addressed most of my comments. A couple of issues persist:

Point 2. In response to a concern I expressed about false discovery (and multiple testing), the authors state that "The outcomes considered in the phenome-wide scan are merely used to provide additional insight on prioritised targets for future follow-up of potential mechanism-based effects on outcomes beyond CHD". I have to say that this explanation doesn't fly. This is somewhat philosophical, but probabilistic tendencies aren't affected by whether one considers a test primary or secondary to ones interests. Accordingly, the burden of testing does include all tests performed, and this needs to be taken into account in any correction procedure. I do however agree that the replication analysis is reassuring.

Response: We thank the reviewer for this comment. We fully agree with the reviewer that a simple classification of primary or secondary analysis, by itself, does not affect multiplicity or false positive rates. However, here we first exclusively focussed on identification of drug targets with a CHD effect and, through independent replication, we reduced the multiple testing burden as the probability of getting two significant results under the null hypothesis became $0.05^2=0.0025$. Only after identifying this subset of CHD prioritized drug targets, did we perform a phenome-wide association (PheWAS) exploring potential additional associations for this subset of prioritised targets, applying the common genome-wide significance threshold of 5×10^{-8} .

Given the order of analyses and its conditional nature, the fact that we explore a phenome-wide scan on the subset of prioritized targets should not inflate the type 1 error rate of the discovery drug target MR analysis for CHD. The false positive rate would of course be inappropriately inflated if we would have performed the drug target MR and PheWAS on all targets, and prioritized results on both analyses.